# MEMORIZATION THROUGH THE LENS OF CURVATURE OF LOSS FUNCTION AROUND SAMPLES

## ABSTRACT

Deep neural networks are over-parameterized and easily overfit the datasets they train on. In the extreme case, it has been shown that these networks can memorize a training set with fully randomized labels. We propose using the curvature of loss function around each training sample, averaged over training epochs, as a measure of memorization of the sample. We use this metric to study the generalization versus memorization properties of different samples in popular image datasets and show that it captures memorization statistics well, both qualitatively and quantitatively. We first show that the high curvature samples visually correspond to long-tailed, mislabeled, or conflicting samples, those that are most likely to be memorized. This analysis helps us find, to the best of our knowledge, a novel failure mode on the CIFAR100 and ImageNet datasets: that of duplicated images with differing labels. Quantitatively, we corroborate the validity of our scores via two methods. First, we validate our scores against an independent and comprehensively calculated baseline, by showing high cosine similarity with the memorization scores released by Feldman & Zhang (2020). Second, we inject corrupted samples which are memorized by the network, and show that these are learned with high curvature. To this end, we synthetically mislabel a random subset of the dataset. We overfit a network to it and show that sorting by curvature yields high AUROC values for identifying the corrupted samples. An added advantage of our method is that it is scalable, as it requires training only a single network as opposed to the thousands trained by the baseline, while capturing the aforementioned failure mode that the baseline fails to identify.

## 1 INTRODUCTION

Deep learning has been hugely successful in many fields. With increasing availability of data and computing capacity, networks are getting larger, growing to billions of parameters. This over-parametrization often results in the problem of overfitting. An extreme form of overfitting was demonstrated by Zhang et al. (2017), who showed that networks can memorize a training set with fully randomized labels. Further, networks make overconfident predictions, even when the predictions are incorrect (Guo et al., 2017), memorizing both mislabeled and long-tailed outliers alike. This can prove to be harmful in real-world settings, such as with data poisoning attacks(Biggio et al., 2012; Chen et al., 2017) and privacy leakage in the form of membership inference attacks (Shokri et al., 2017). There has been considerable research effort towards countering overfitting, with different forms of regularization (Krogh & Hertz, 1991), dropout (Srivastava et al., 2014), early stopping (Yao et al., 2007) and data augmentation (DeVries & Taylor, 2017; Zhang et al., 2018). In this paper, we exploit overfitting to propose a new metric for measuring memorization of a data point, that of the curvature of the network loss around a data point. We overfit intentionally and utilize this to study the samples the network is memorizing.

We focus our analysis on examples with excessive loss curvature around them compared to other training data samples. We find that such samples with high curvature are rare instances, i.e. drawn from the tail of a long-tailed distribution (Feldman, 2020; Feldman & Zhang, 2020), have conflicting properties to their labels or are associated with other labels, such as multiple objects, and most significantly, mislabeled samples. Figure 1 shows cherry-picked examples from the 100 highest curvature samples from different vision datasets. Consider the images shown for CIFAR100, categories such as 'girl' and 'baby' and 'folding_chair' and 'rocking_chair' from ImageNet. The images are

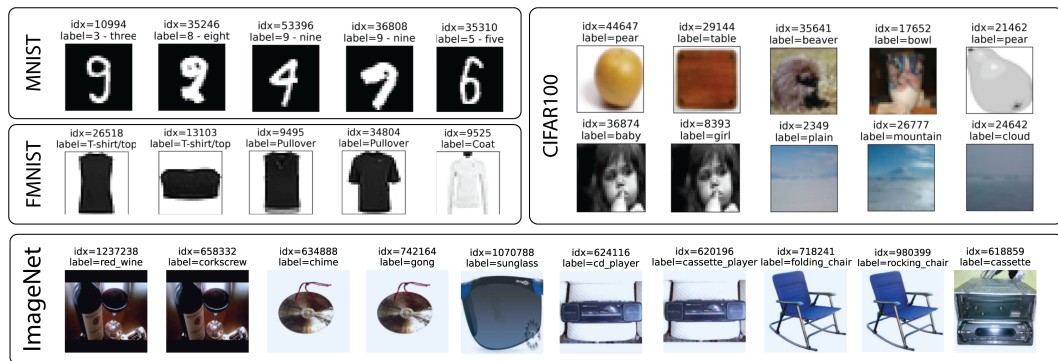

Figure 1: Cherry picked examples from the 100 highest curvature training examples on MNIST, FMNIST, CIFAR100 and ImageNet.

the same, and they semantically belong to both categories, but would be confusing for single label datasets. Similarly in CIFAR100, images for the classes of 'plain', 'mountain', and 'cloud' have a large overlap. Such similar or overlapping samples can be challenging to catch while creating datasets, and can then get allotted to separate annotators that label them differently. ImageNet labels are further complicated due to multi-object images as is the case with 'red_wine' and 'corkscrew' where both the objects are visible in the image, hence creating conflicting but semantically correct labels. Further, when looking at high curvature ImageNet images we also find other failure modes, such as a non-prototypical image for 'sunglasses', mislabelled examples of 'cassette' when clearly it is a 'cassette_player', and conflicting conflicting labels for 'chime' and 'gong'.

The quality of datasets impacts not only the accuracy of the network but also encodes hidden assumptions into the datasets, raising issues of bias and unfairness. Underrepresented samples of a class in a dataset can have a significant impact on the performance of the network in real-life settings (Mehrabi et al., 2021). For datasets not vetted by the public, or those that are weakly labeled or have noisy annotations, curvature analysis can help to audit datasets and improve their quality. As an example, while studying curvature properties of CIFAR100 and ImageNet vision datasets we find that nearly half of the top 100 high curvature images correspond to duplicated images *with different labels* in their training sets (Figure 4 and Appendix Figure 13). While the duplication of images in CIFAR100 has been noted before by Recht et al. (2019) and Barz & Denzler (2020), differently labeled duplicated images are, to the best of our knowledge, a novel observation.

In order to validate our measure of curvature quantitatively, we show that our scores achieve a high cosine similarity with memorization scores released by Feldman & Zhang (2020) for CIFAR100 and ImageNet datasets. The baseline scores were calculated from extensive experimentation, and required training thousands of models. In contrast, we can calculate curvature while training only one network, while identifying failure modes that the baseline fails to capture. As a second, independent check, we synthetically mislabel a small proportion $(1 - 10\%)$ of the dataset by introducing uniform noise in the labels and overfit a network to this noisy dataset. We then use our method to calculate the AUROC score for identifying the mislabeled examples. We consistently get high AUROC numbers for the datasets studied, highlighting that the mislabeled samples were learned with high curvature.

## 2 RELATED WORK

Deep neural networks are highly overparametrized, thus suffer from overfitting. Overfitting leads networks to memorize training data to perfect accuracy, while not improving generalization capability. There has been work showing that overfitting in neural networks might be benign, and might not harm testing accuracy (Bartlett et al., 2020). However, overfitting can create other problems such as increased susceptibility to membership inference attacks (Shokri et al., 2017; Carlini et al., 2019b), and compromised adversarial robustness (Rice et al., 2020). The extremity of memorization was first noted in neural networks by Zhang et al. (2017). Since then, there has been work in understanding memorization better (Arpit et al., 2017; Feldman & Zhang, 2020; Feldman, 2020; Jiang et al., 2021; Stephenson et al., 2021). Feldman & Zhang (2020) suggest calculating memorization score

by removing a set of examples from the training set and observing the change in their prediction upon training with and without the samples. Related to memorization, research on the easiness or hardness of examples by Johnson & Guestrin (2018) and Katharopoulos & Fleuret (2018) utilize the gradient of the sample as the metric of its importance. AUM (Pleiss et al., 2020) considers the difference between the target logit and the next best logit as a measure of importance, and Maini et al. (2022) captures the sample importance in the time it takes to be forgotten when the network is trained with the sample removed. Carlini et al. (2019a) study the prototypicality and memorization of samples based on 5 different correlated metrics.

Methods to counter overfitting include weight decay penalty (Krogh & Hertz, 1991), dropout (Srivastava et al., 2014) and augmentation (Shorten & Khoshgoftaar, 2019). However, the success of these techniques depends on having a clean and reliable training dataset. Confident Learning (Northcutt et al., 2021a) uses the principles of pruning, ranking and counting with out of prediction probabilities to find mislabeled data. The authors extend it (Northcutt et al., 2021b) to show errors in many common image datasets. Recht et al. (2019) build new test sets for CIFAR and ImageNet to check if there has been unintentional overfitting to the released validation sets. Barz & Denzler (2020) find many duplicated samples in the CIFAR dataset and create a tool to flag possible duplicates. We looked at the worst-case examples from these methods and found that they did not outrightly catch duplicated examples with different labels, whereas our method does. The curvature of the loss function *with respect to the parameters* is well studied in its connection to the generalization capability of the solution (Keskar et al., 2016; Dinh et al., 2017; Ghorbani et al., 2019). However, far fewer works focus on the properties of curvature of loss *with respect to data*. Prior work in this area has focused on adversarial robustness (Moosavi-Dezfooli et al., 2019; Fawzi et al., 2018), and on coresets (Garg & Roy, 2023). While prior work has focused on adversarial robustness, in this paper, we propose a new application of input loss curvature as a metric for memorization.

## 3 METHODOLOGY

In this section, we outline how we measure curvature and the computational cost involved. A similar method is often used to calculate the curvature of loss with respect to the network parameters to determine solution stability (Dinh et al., 2017; Ghorbani et al., 2019; Keskar et al., 2016). However, we use this method to calculate the curvature of the loss with respect to the input points. It is conceptually similar to the method used for curvature regularization in Moosavi-Dezfooli et al. (2019) and Garg & Roy (2023) with differences in curvature calculation (see Appendix D.4) and hyperparameters used. Details regarding the hyperparameters are provided in Appendix B.

### 3.1 MEASURING CURVATURE

Let $X \in \mathbb{R}^D$ be the input to a neural network. Let $y \in \{1, 2, .., C\}$ be the assigned label for this point, corresponding to the index of the true class among $C$ classes. Let $\hat{y}_t = f(X, W_t) \in \mathbb{R}^C$ be the output (pre-softmax logits) of the network with weights $W_t$ at epoch $t$. Let $L_t(X) = CrossEntropy(\hat{y}, y) \in \mathbb{R}$ be the loss of the network on this data point. We are interested in the Hessian of the loss with respect to $X$, $H(X) \in \mathbb{R}^{D \times D}$, where each element of the matrix is defined by

$$[H(X)]_{i,j} = \left[ \frac{\partial^2 L(X)}{\partial x_i \partial x_j} \right]; i, j = 1, 2..., D \tag{1}$$

Henceforth, we refer to $H(X)$ as $H$ from now, implicitly understanding that it is calculated concerning datapoint $X$ at a given weight $W$. The local curvature is determined by the eigenvalues of $H$ (Dinh et al., 2017; Ghorbani et al., 2019; Keskar et al., 2016). The sum of the eigenvalues is also the trace of the $H$, and can be calculated using Hutchinson's trace estimator (Hutchinson, 1990).

$$Tr(H) = \mathbb{E}_v \left[ v^T H v \right] \tag{2}$$

where $v \in \mathbb{R}^D$ belongs to a Rademacher distribution, i.e $v_i = \{+1, -1\}$ with equal probability. However, we are more interested in the magnitude of the curvature rather than the definiteness, and hence we look at the trace of the square of the hessian, which computes to the sum of the square of

the eigenvalues. Since the Hessian is symmetric, we have:

$$
\begin{aligned}
Tr(H^2) &= \mathbb{E}_v \left[ v^T H^2 v \right] \\
&= \mathbb{E}_v \left[ (Hv)^T (Hv) \right] \\
&= \mathbb{E}_v \| Hv \|_2^2 \\
&= \frac{1}{n} \sum_{i=0}^{n} \| Hv_i \|_2^2
\end{aligned}
\tag{3}
$$

where $n$ is the number of Rademacher vectors to average over. Similar to Moosavi-Dezfooli et al. (2019) and Garg & Roy (2023), we use finite step approximation to calculate this efficiently.

$$
Hv \approx \frac{1}{h} \left[ \frac{\partial L(x + hv)}{\partial x} - \frac{\partial L(x)}{\partial x} \right]
\tag{4}
$$

$$
Hv \propto \frac{\partial \left( L(x + hv) - L(x) \right)}{\partial x}
\tag{5}
$$

We drop constants as we are only interested in the relative curvatures of datapoints. For our final curvature estimate, we average curvature over all training epochs, T, to give reliable results. Putting this together, we have the curvature estimator of a datapoint $X$ at any epoch as:

$$
Curv(X) = \frac{1}{nT} \sum_{t=1}^{T} \sum_{i=0}^{n} \left\| \frac{\partial (L_t(x + hv) - L_t(x))}{\partial x} \right\|_2^2
\tag{6}
$$

### 3.2 COMPUTATIONAL COST

The computational complexity increases from $O(T)$ for vanilla training to $O(nT)$ for calculating curvature scores, where $n$ is the number of Randemacher vector $v$ to average and $T$ is the number of epochs. The backward passes can be parallelized as they are performed at a static $W$. In case of limited compute capacity, curvature could be estimated every few epochs. Note that we do not explicitly compute the Hessian or any second-order gradient despite relying on a second-order metric.

### 4 EXPERIMENTS AND DISCUSSION

In this section, we present qualitative and quantitative results on MNIST (Deng, 2012), Fashion-MNIST (Xiao et al., 2017), CIFAR10/100 (Krizhevsky et al., 2009), and ImageNet (Russakovsky et al., 2015) datasets to support our claim that curvature can be used to measure memorization. The histograms of curvatures are shown in Figure 2, and we can see that they reflect the long-tailed nature of natural images. We use ResNet18 (He et al., 2016) for all our experiments. Curvature for all datasets except ImageNet is calculated every epoch and averaged. For ImageNet we average curvature calculated every 4 epochs for computational ease. Details of the setup are given in Appendix C. This section first shows qualitative results by visualizing high curvature samples, and showing that they are long-tailed or mislabeled images in section 4.1. We then validate our score quantitatively in section 4.2 by comparing against a baseline and in section 4.3 by synthetically mislabeling data and showing that it gets learned with high curvature. Lastly, in section 4.4, we study how our measure of curvature evolves during training for further insight, and so that practitioners with limited compute can better choose limited epochs for curvature calculation.

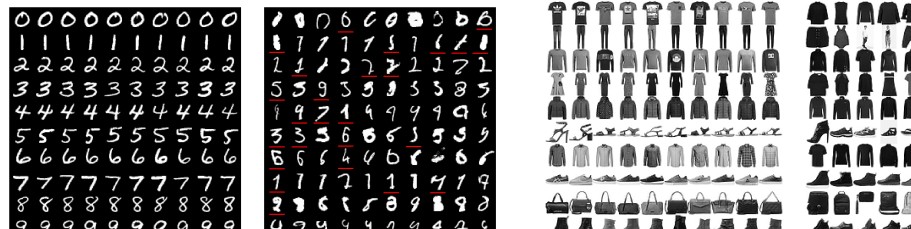

Figure 3: Low (left) and high (right) curvature samples of MNIST and FashionMNIST training data

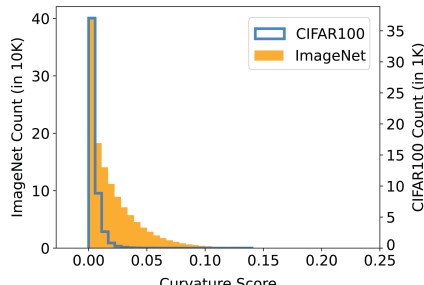

Figure 2: Histogram of curvature scores

Table 1: Cosine Similarity (CS) between curvature and FZ scores with and without weight decay (WD). Top-K CS is the CS of the top 5,000 (and 50,000) FZ score samples of CIFAR100 (and ImageNet).

| Dataset | WD | CS | Top-K CS |
|---------|-----|------|----------|
| CIFAR100 | 1e-04 | 0.82 | 0.90 |
|          | 0.0 | 0.73 | 0.82 |
| ImageNet | 1e-04 | 0.72 | 0.87 |
|          | 0.0 | 0.66 | 0.82 |

## 4.1 VISUALIZING LOW AND HIGH CURVATURE SAMPLES

In Figure 3, we visualize the 10 lowest and highest curvature training samples for each class of MNIST and FMNIST respectively. These are examples sorted by curvature scores averaged over all training epochs of a modified ResNet18 architecture, trained without weight decay and allowed to be fully memorized (100% training accuracy). We see that low curvature samples appear to be easily recognizable as belonging to their class. Whereas, high curvature samples, are made of both long-tailed (harder samples or rare occurrences) and mislabeled samples. The mislabeled samples in MNIST are underlined in red, and we note that the remaining high curvature samples are also ambiguous. In particular, we see high overlap between classes '4' and '9', and '1' and '7'.

The classes for FashionMNIST are 0:T-shirt/top, 1:Trouser, 2:Pullover, 3:Dress, 4:Coat, 5:Sandal, 6:Shirt, 7:Sneaker, 8:Bag, 9:Ankle boot. We see that examples seem to have significant overlap for classes 'T-shirt', 'Pullover' and 'Coat', and among 'Ankle boots' and 'Sneakers'. We also see long-tailed trends in the row of 'handbags', with some uncommon handbags being represented in the high curvature case. Conflicting samples are apparent in the row for pants, as there are full body pictures of models, wearing both pants and shirts. This analysis reveals a bias in the FashionMNIST dataset: darker clothes with lower contrast tend to show up as high curvature samples, whereas in the real world, this may not be the case. The corresponding pictures for the training sets of CIFAR10, CIFAR100 and ImageNet are shown in Appendix D. We check the validation data as well by training and overfitting on the validation sets, and the results are shown in Appendix E.

## 4.2 COMPARING AGAINST A BASELINE: FZ SCORES

Here, we are interested in an independent measure of memorization that does not utilize training dynamics. The most suitable metric for comparison comes from Feldman & Zhang (2020), who remove datapoints from the dataset one at a time, train a new network on the altered dataset, and measure the change in prediction on the datapoint as the sample memorization score. These scores are likely to be independent of spurious correlations to curvature that other methods such as confidence of prediction might have, and hence serve as a good baseline. The authors released scores for the CIFAR100 and ImageNet dataset on their site[1], and we refer to these as FZ scores. We calculate the cosine similarity of our curvature with the FZ scores in Table 1. **We achieve a cosine similarity of 0.82 and 0.72 for CIFAR100 and ImageNet respectively when we use weight decay, and 0.90**

---

[1]https://pluskid.github.io/influence-memorization/

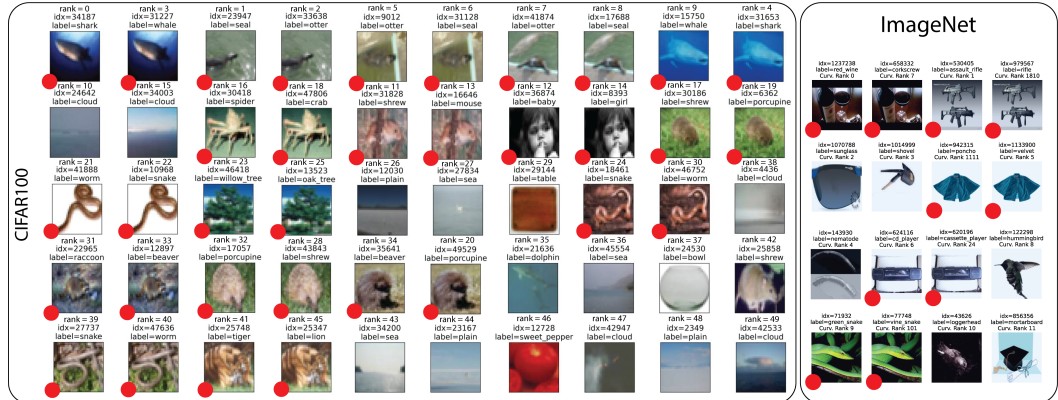

Figure 4: 50 highest curvature samples from the training set of CIFAR100 and ImageNet, index, label (and curvature rank for ImageNet) shown above each image. We highlight duplicated samples with differing labels with a red dot. Extended and ordered versions in Appendix, Figures 11 and 13.

**and 0.87 on the most memorized samples** (see Table 1 and Appendix D.5 for additional results). These scores are a high match since the vector is of dimension 50,000 in the case of CIFAR100, and $1,268,356$ for ImageNet (number of training samples in the dataset). While the dataset level cosine similarity suggests correlations in the aggregate, it does not provide specific sample-based correlations. However, the significant match between FZ and curvature scores does suggest a very strong link. Further, we emphasize that we achieve this match with only 1 training run, whereas FZ scores required training thousands of models.

Counterintuitively, we note that the cosine similarity drops by $\sim 0.1$ when not using weight decay. To understand why, we visualize the examples with the highest curvature without weight decay in Figure 4 for CIFAR 100 and ImageNet. We note that **36 out of the 60 highest curvature samples for CIFAR100, and 45 out of top 100 for ImageNet are duplicated pairs with conflicting labels.** These are marked with a red dot. In contrast, FZ scores that have been released do fail to catch duplicate samples with different labels. Similarly, weight-regularized, highest curvature samples only catch a few of the duplicated samples, as regularization would deter the network from memorizing these samples. The 100 most memorized samples from FZ scores and the highest curvature samples for training with weight decay are shown in Appendix F. These duplicated samples are indeed memorized during training but possibly missed by FZ scores due to the fact that they do not train until complete memorization due to the computational expense of training thousands of models. Additionally, FZ scores are calculated by removing a proportion of the dataset at a time and training on the remaining samples. Duplicates can compromise the reliability of these scores. We note that there are many other duplicate samples that our method does not identify (Barz & Denzler, 2020), since these duplicate samples have the same label and do not pose a boundary conflict. They are also not likely to be memorized.

To summarize, one of the reasons we do not get near perfect match with FZ scores (and why we get a higher match with weight decay) is because **we catch a novel failure mode that the FZ scores fail to, despite being** $\sim 3$ **orders of magnitude more computationally expensive.** For the sake of completeness, we recommend that practitioners try both settings (with and without weight decay) to try and catch the different kinds of boundary conflicts that may be revealed via curvature analysis. We do a more detailed analysis in section 4.4, where we show the cosine matches per epoch.

### 4.3 SYNTHETIC LABEL CORRUPTION

To provide further evidence of curvature as a memorization metric, we devise an experiment to measure how well our method captures synthetically mislabeled examples, since they are most likely to be memorized. We randomly introduce noise into a proportion of the labels, uniformly changing its class label to a different class label. The same proportion of labels in each class are corrupted. We then train on these samples and measure the curvature scores. First, we randomly corrupt the label of 60 images per class of MNIST (corrupting 1% of the dataset), and train until full memorization, i.e. 100% training accuracy, indicating that the mislabeled samples were memorized. We plot the

Table 2: AUROC for identifying corrupted samples with synthetic label noise, best results are shown in red. Inconf. is inconfidence score, LT and SSFT are learning time and second split forgetting time from Maini et al. (2022) and CL refers to Confident Learning (Northcutt et al., 2021a).

| Dataset | Method | Corruption | | | | | |
|---------|--------|------|------|------|------|------|------|
| | | 1% | 2% | 4% | 6% | 8% | 10% |
| MNIST | Inconf. | 99.4% | 99.0% | 98.4% | 97.7% | 97.1% | 96.1% |
| | CL | 99.7% | 99.3% | 99.1% | 98.9% | 98.9% | 98.9% |
| | SSFT | 99.9% | 99.9% | 99.8% | 99.7% | 99.7% | 99.5% |
| | LT | 98.7% | 99.5% | 98.4% | 98.5% | 97.7% | 97.3% |
| | Curv | **100.0%** | **100.0%** | **99.9%** | **99.9%** | **99.9%** | **99.9%** |
| CIFAR10 | Inconf. | 84.2% | 82.5% | 81.8% | 81.6% | 81.5% | 81.5% |
| | CL | 92.4% | 93.1% | 93.4% | 93.4% | 91.7% | 93.4% |
| | SSFT | 94.5% | 94.1% | 93.2% | 92.5% | 91.6% | 90.0% |
| | LT | 87.3% | 82.5% | 84.0% | 83.4% | 82.2% | 83.0% |
| | Curv | 97.4% | 96.6% | 95.5% | 94.4% | 94.1% | 92.9% |
| | Curv$^{\text{SSFT}}$ | **97.5%** | **96.8%** | **96.2%** | **96.1%** | **95.2%** | **94.8%** |
| CIFAR100 | Inconf. | 85.0% | 84.0% | 83.5% | 83.6% | 83.6% | 83.4% |
| | CL | 83.1% | 84.2% | 85.3% | 86.3% | 84.3% | 84.6% |
| | SSFT | 94.8% | 93.7% | 93.2% | 91.7% | 91.9% | 91.0% |
| | LT | 85.2% | 85.2% | 81.8% | 80.8% | 79.8% | 78.5% |
| | Curv | 90.9% | 89.6% | 88.3% | 86.8% | 85.7% | 84.3% |
| | Curv$^{\text{SSFT}}$ | **96.0%** | **94.5%** | **94.2%** | **93.1%** | **92.9%** | **91.9%** |

histogram of the cumulative curvature scores and mark the scores of the corrupted ones with a black × in Figure 5. The training dataset consists of 60,000 samples and we can see that most of the samples have very low curvature. However, we see that **the curvature scores of the corrupted examples are among the highest,** confirming the link between curvature and memorization.

To do a more exhaustive study, we sort samples by their curvature estimate and report the AUROC (Area under the ROC curve) scores for separating the corrupted samples from the clean ones in Table 2, for corruptions ranging between 1-10%. The best results are highlighted in red. As shown later while studying curvature dynamics in section 4.4, the scores averaged over training are stable when using weight decay, and hence we use a weight decay of $10^{-4}$ for all models for these experiments. For additional baselines, we also provide AUROC results using inconfidence (Carlini et al., 2019a) and Confident Learning (Northcutt et al., 2021a). We also show the results of Leanring Time (LT) and Second-Split Forgetting Time (SSFT) from Maini et al. (2022). The authors of SSFT note in their text that SSFT can be combined with other methodologies, and we show the results of using their method combined with our curvature score as Curv$^{\text{SSFT}}$. The experiments for CL, LT and SSFT were performed using the code provided by respective authors.
From Table 2, we conclude that our method is more informative than just confidence values on all datasets and outperforms CL and LT significantly as well on all corruptions and datasets considered. We also outperform SSFT on all corruptions considered for both MNIST and CIFAR10. However, we note that SSFT shows better performance on CIFAR100, but combining SSFT and Curvature outperforms either method significantly. From Table 2 we can see that Curvature outperforms SSFT for

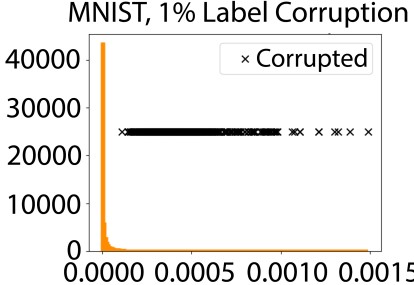

Figure 5: Histogram of MNIST trainset curvature. Curvature of 60 corrupted samples marked.

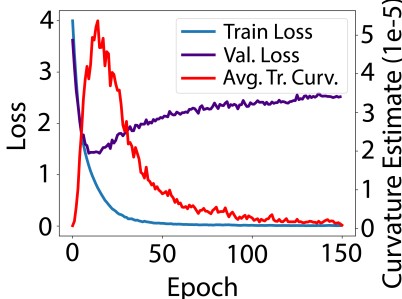

Figure 6: Loss curves and average trainset curvature estimates on CIFAR100.

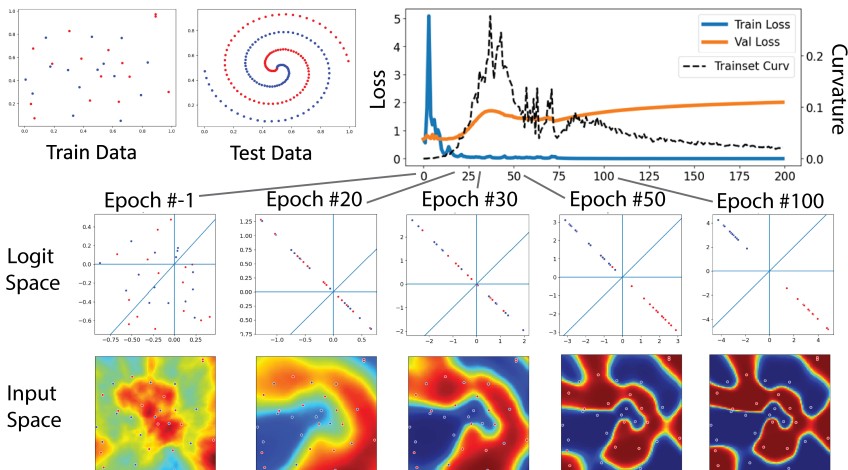

Figure 7: Visualization of curvature of decision boundary around data points. Train and test datasets are shown on the top left, and loss curves on the top right. The bottom figures show the logit space and the decision boundary with input points in the input space at different epochs of training.

both MNIST and CIFAR-10 for all percentages of mislabeling considered. However, SSFT performs better than Curvature for CIFAR-100, but as We show that combining SSFT and Curvature results in a combination that outperforms both SSFT and Curvature independently and significantly. We emphasize here that we do not claim that curvature is the best way of finding mislabeled examples. Our primary motivation is to show that curvature is a good measure of sample memorization, and it can find samples with failure modes that other methods might miss. We recommend that curvature analysis should be used in conjunction with other checks, for a holistic view of dataset integrity.

## 4.4 CURVATURE DYNAMICS DURING TRAINING

We now study per-epoch curvature, to explain why we need to average curvature scores over training epochs. Figure 6 plots the training and validation losses and the per-epoch curvature score averaged for the training set of CIFAR100 on ResNet18. The network begins to overfit around epoch 20, and the per-epoch curvature exhibits a peculiar trend: it increases until overfitting takes hold, then decreases. This trend is also observed on ImageNet (Figure 8b).

To investigate this behaviour, we created a 2D toy example (training and setup details in Appendix A). Figure 7 shows the training and test data, consisting of 2 classes (red and blue). The test data is abundant and noise-free, representing the true distribution. The training data consists of only 15 points for each class, with 30% corrupted by noise. The imbalance in the design of this toy example is intentional. Our goal was to evaluate what the decision boundary looks under extreme memorization. The dynamics of training are captured in Figure 7 which shows the loss curves for training and testing data (cross-entropy loss). Snapshots of the output space (logits) and input space are shown at the bottom. The output space shows the $x$ and $y$ axes, corresponding to each class. The decision boundary in the output space is the line $y = x$. The input space visualizes the probability heatmap of each point in the grid belonging to the red class. This setup allowed us to understand how curvature behaves with extreme memorization.

First we note what the boundaries look like in the **input space**. From snapshots at epochs 20, 30 and 50, we see that the network tries out different hypotheses in the form of different decision boundaries. This is even more evident if we look at the epoch-wise GIF created from these snapshots in the supplementary material. **Averaging the scores over all epochs, allows us to not be overly reliant on one hypothesis, and account for all decision boundary considered during learning.** Secondly, we note that the loss and curvature plots show the same trend of increasing and then decreasing in this toy example as well. This can be explained by looking at the **logit space** when using the cross-entropy loss. Early in training, the network aims to get all data points to the correct side of the boundary. This phase is marked by a very large sensitivity of loss to perturbation of the input, resulting in large values of curvature. As learning progresses, the logits get further apart and the

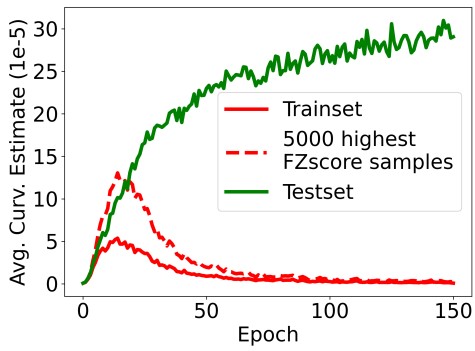 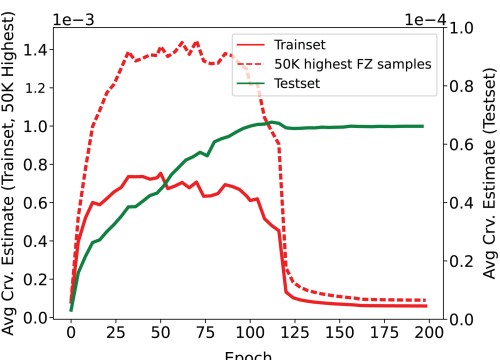

(a) Average trainset, testset and memorized trainset curvatures on CIFAR100, caclulated every epoch.

(b) Average trainset, testset and memorized trainset curvatures on ImageNet, calculated every 4 epochs.

Figure 8: Curvature dynamics for CIFAR100 and ImageNet.

softmax layer becomes more confident. Even when they are all on the right side, the cross entropy keeps maximizing the margin, by pushing logits further and further apart from the boundary. This means that small perturbations have little impact on the softmax output and the gradients, which are used to measure curvature. Hence, the curvature values drop.

Note this trend does not affect the results discussed in sections 4.1, 4.2, and 4.3 as they are averaged over epochs which allows us to be impervious to these dynamics. Interestingly, the curvature of validation samples increases throughout training. This is because the margin for test samples has not been maximized (since these were not trained on). And this supports the fact that early stopping helps lower adversarial vulnerability (Rice et al., 2020) and longer training makes the model more susceptible to membership inference attacks (Shokri et al., 2017).

To re-emphasize the need for averaging over epochs, we show the cosine similarity match between per-epoch curvature scores and FZ scores, along with the cosine similarity between the cumulative curvature scores and FZ scores in the Appendix (Section D.4 and Figures 14b, 14a and 15). We see that before overfitting, the cosine match between *per-epoch* curvature and FZ score is the same as the cosine match between *cumulative* curvature and FZ score. The match drops for per-epoch case and when not using weight decay, as cross entropy forces logit separation and margin maximization at the later epochs. This effect is more aggressive without weight decay regularization, as can be noted from the plots. Hence, averaging over all the epcochs gives us stable and reliable results.

## 5 CONCLUSION

Overparametrized networks are known to overfit to the training dataset, and can achieve 100% training accuracy by memorizing training datapoints. Zhang et al. (2017) demonstrate an extreme version of this, wherein they show that the networks can fully memorize a training dataset with all labels randomized to perfect accuracy. This raises concerns about the networks consuming erroneous data, which can have an adverse impact when the decisions made by neural networks are used in real-life scenarios. In this paper, we propose curvature of the loss function around the datapoint, measured as the trace of the square of the Hessian, as a metric of memorization. We overfit to the training set and measure the curvature of the loss around each sample. We validate curvature as a good measure of memorization in three ways. First, we visualize the highest curvature samples and note that they are made of mislabeled, long-tailed, multiple class, or conflicting samples that are not clearly representative of their labels. Second, We also show that curvature estimates have a very high cosine similarity match with FZ scores, which are calculated by training thousands of models per dataset. Instead, our method only requires training one network. Using our method, we catch a failure mode on CIFAR100 and ImageNet that is to the best of our knowledge, unobserved until now; that of duplicated images with different labels. Third, we show that using curvature to identify mislabeled samples in the case of synthetically mislabeled training sets achieves high AUROC scores. These three experiments help us establish curvature as a reliable and scalable method for measuring memorization of a sample by a network. This can be utilized to check the integrity of datasets and identify undersampled or baddly annotated parts of the dataset. Finally, we study how curvature develops during training, giving us insight into the dynamics of overfitting.

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

## A   2D TOY EXAMPLE VISUALIZATION

We use a network of 7 fully connected layers of size [2,100,100,500,200,100,1000]. Each layer is followed by a batchnorm layer and ReLU layer. The network is trained for 150 epochs with an SGD optimizer and a learning rate of 0.1, momentum of 0.9 and a weight decay of 5e-4. There is only one minibatch per epoch. The training dataset consists of 15 points for each of the 2 classes, with a noise ratio of 0.3 introduced to 30% of the data. The test dataset consists of 100 points for each class with no noise. In Figure 7, at the bottom we show the input space, specifically, we visualize the probability heatmap of each point in the grid belonging to the red class. And we also add the visualization of the output space. Early epochs show large changes in the curvature around points in the input space. We can see from the output space that the focus of early epochs is to get all datapoints on the right side of the boundary. Once the datapoints are on the right side, the cross entropy loss still aims to maximize the log confidence of the correct class, by maximally separating the correct class logit from the others. In these epochs, while the logits get more separated, little change is seen in the input space. This leads to decreasing curvature after overfitting.

## B   HYPERPARAMETERS

Our estimator introduces two hyperparameters, $h$ and $n$. We tune h in the range of $\left[10^{-2}, 10^{-4}\right]$ by evaluating the cosine similarity between our scores and FZ scores. We note that input $X$ is in the range of $[0, 1]$ (before mean-standard normalization) and all elements of $v \in \{+1, -1\}$. This means that the $L_2$ distance of the noise added to the input remains in the range of $||hv||_2 = h\sqrt{D}$, or equivalently, we add perturb each pixel by $\pm h$. We report the result of tuning hyperparameters in Table 3.

| h | n | Top-K CS | CS |
|---|---|---|---|
| 0.01 | 5 | 0.89 | 0.80 |
| 0.001 | 5 | 0.90 | 0.82 |
| 0.0001 | 5 | 0.90 | 0.82 |
| 0.01 | 10 | 0.89 | 0.80 |
| 0.001 | 10 | 0.90 | 0.82 |
| 0.0001 | 10 | 0.90 | 0.82 |
| 0.01 | 20 | 0.89 | 0.80 |
| 0.001 | 20 | 0.90 | 0.82 |
| 0.0001 | 20 | 0.90 | 0.82 |

Table 3: Cosine Similarity (CS) between curvature and FZ scores with weight decay (WD). Top-K CS is the CS of the top 5,000 FZ score samples of CIFAR100.

## C    NETWORK ARCHITECTURE AND TRAINING DETAILS FOR CURVATURE ESTIMATES

We use modified versions of ResNet18 for all experiments with appropriately modified input sizes and channels. For MNIST and FashionMNIST, we use the downscaled ResNet [2] with the average pooling layer downsized from 8 to 7 due to the reduced input resolution of MNIST and FashionM-NIST. For CIFAR datasets we use the full-size ResNet18[3]. WE use PyTorch provided ResNet18 for ImageNet models. We use no augmentation for MNIST and FashionMNIST, and random horizontal flips and crops for CIFAR and ImageNet datasets. We train for 300 epochs on CIFAR datasets, with a learning rate of 0.1, scaled by 0.1 on the $150^{th}$ and $250^{th}$ epoch. For MNIST and FashionMNIST, we train for 200 epochs, with a learning rate of 0.1 scaled by 0.1 on the $80^{th}$ and $160^{th}$ epoch. For ImageNet we train for 200 epochs with a learning rate of 0.1, scaled by 0.1 on the $120^{th}$ and $160^{th}$ epoch. Where weight decay is used, its value is set to $10^{-4}$.

## D    TRAINING SAMPLES WITH LOW AND HIGH CURVATURE

### D.1    CIFAR10

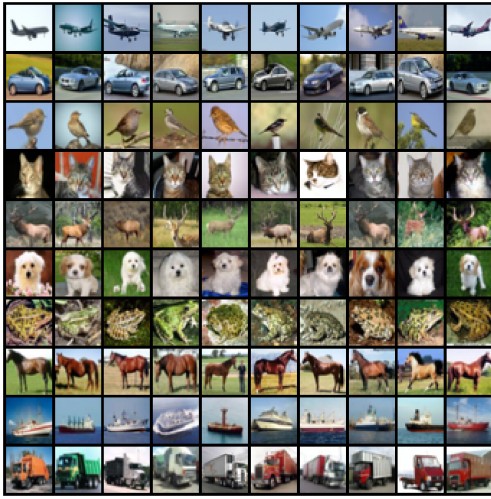 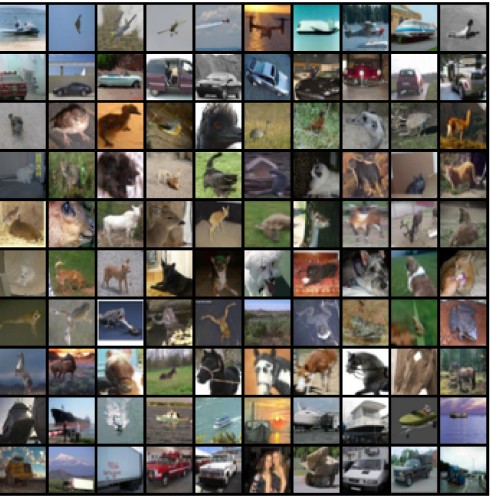

Figure 9: Low and High curvature samples from CIFAR10. Classes from top to bottom: [airplanes, cars, birds, cats, deer, dogs, frogs, horses, ships, and trucks]

[2]https://github.com/bearpaw/pytorch-classification/blob/master/models/cifar/resnet.py
[3]https://github.com/kuangliu/pytorch-cifar/blob/master/models/resnet.py

The low curvature samples are shown on the left, with the high curvature samples on the right. The classes in order are: [airplanes, cars, birds, cats, deer, dogs, frogs, horses, ships, and trucks].

## D.2   CIFAR100

The low curvature samples are shown on the left, with the high curvature samples on the right. We show samples only from the first 10 classes, with the following labels: [apple, aquarium fish, baby, bear, beaver, bed, bee, beetle, bicycle, bottle].

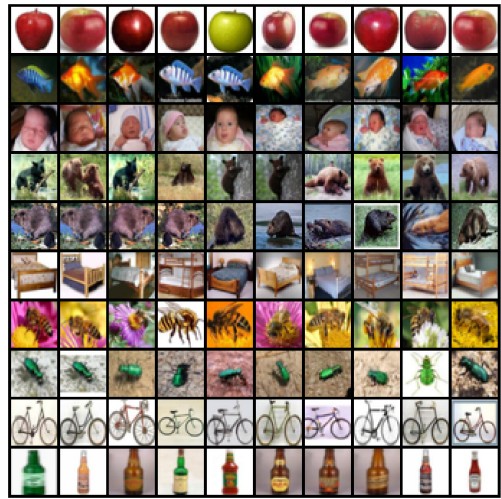 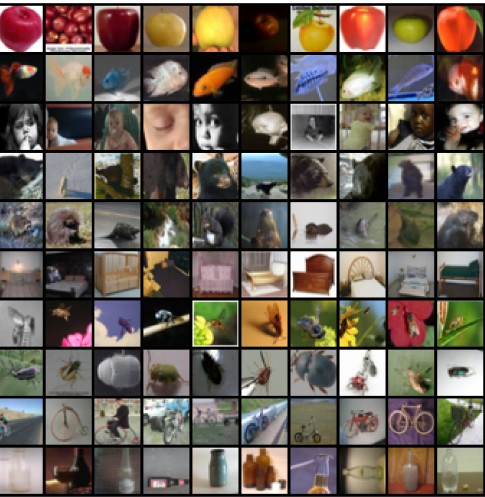

Figure 10: Low and High curvature samples from CIFAR100. Classes from top to bottom: [apple, aquarium fish, baby, bear, beaver, bed, bee, beetle, bicycle, bottle].

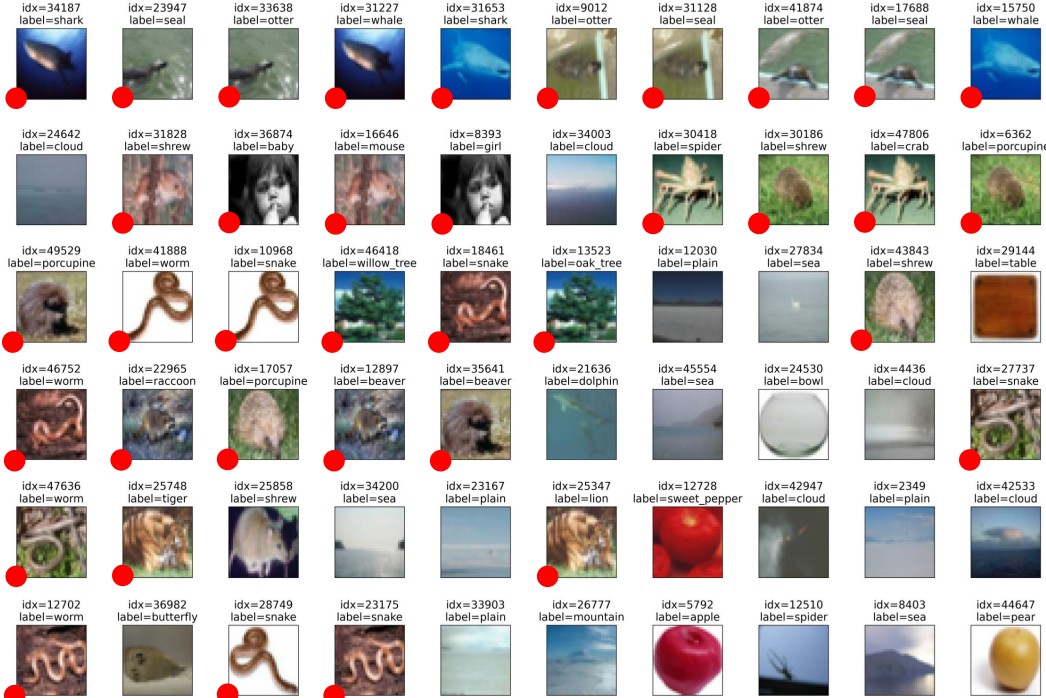

Figure 11: 60 highest curvature samples from the training set of CIFAR100, identified by training on ResNet18. The index and label of the training sample are mentioned above the picture. We highlight samples that are duplicated with differing labels with a red dot.

### D.3 IMAGENET

For ImageNet we show samples only from the first 10 classes, with the following labels: [tench, goldfish, great white shark, tiger shark, hammerhead, electric ray, stingray, cock, hen, ostrich]. Note that low curvature samples on 'tench' on ImageNet reveal a spurious correlation learnt by the network between tench and people holding the 'tench' fish.

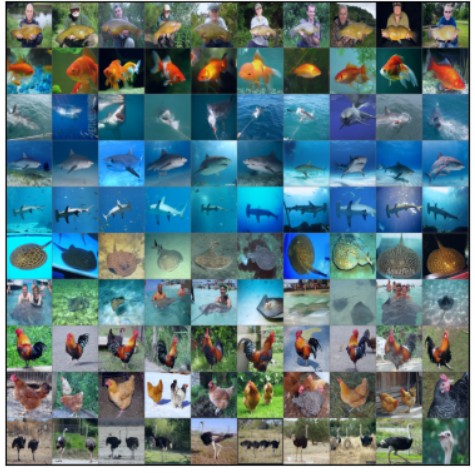 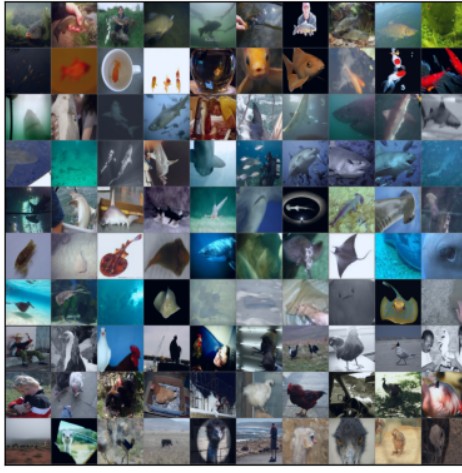

(a) Low curvature samples from ImageNet    (b) High curvature samples from ImageNet

Figure 12: Low and High curvature samples from ImageNet. Classes from top to bottom: [tench, goldfish, great white shark, tiger shark, hammerhead, electric ray, stingray, cock, hen, ostrich].

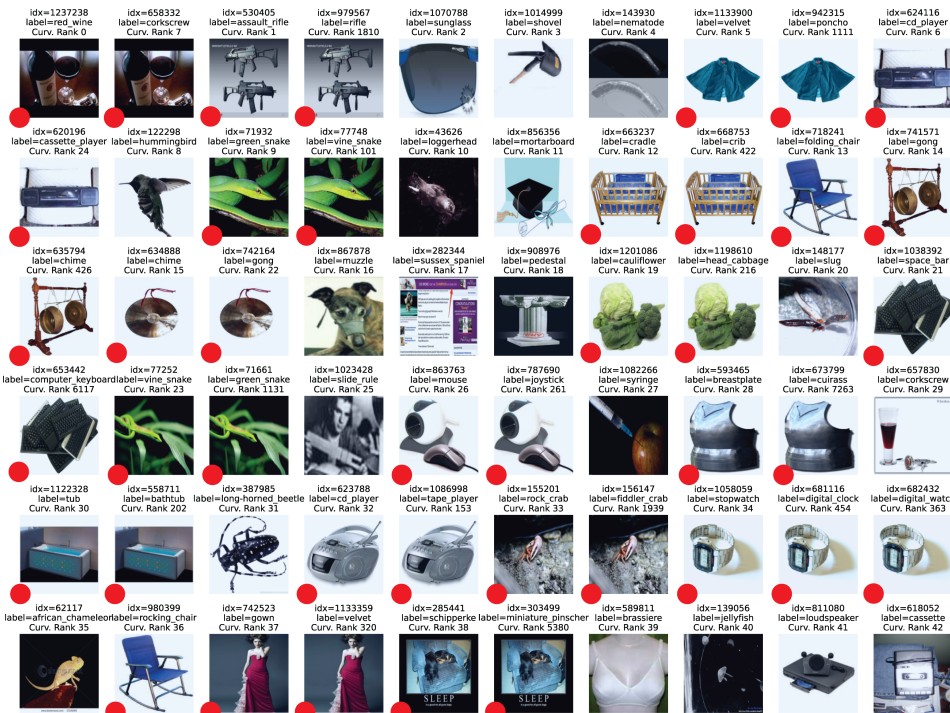

Figure 13: Top curvature samples from the training set of ImageNet along with duplicates with corresponding curvature ranks. Scores identified by training on ResNet18. We highlight samples that are duplicated with differing labels with a red dot.

## D.4 EPOCH-WISE COSINE SIMILARITY

We plot the epoch-wise and cumulative cosine similarity between the FZ score and curvature scores for CIFAR100 (Figure 14b) and ImageNet (Figure 14a). Further, we also plot the epoch-wise and cumulative cosine similarity between top 50K FZ sample memorization scores and the corresponding curvature scores for ImageNet (Figure 15). We get very high similarity $\sim 0.9$

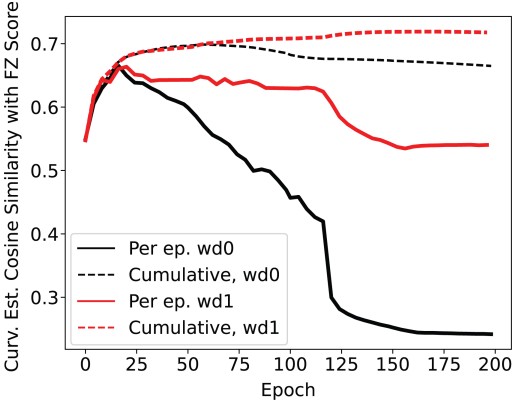

(a) Cumulative and epoch-wise curvature of the training set of ImageNet with and without weight decay.

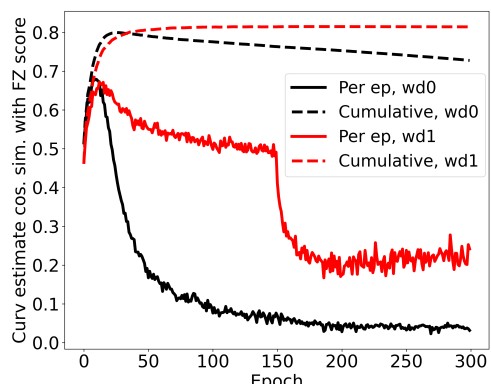

(b) Cumulative and epoch-wise curvature of the training set of CIFAR100, with and without weight decay

Figure 14: Cosine similarity between curvature and FZ Memorization score for ResNet18 on ImageNet (left) and CIFAR100 (right). ImageNet results are plotted every 4 epochs for efficiency.

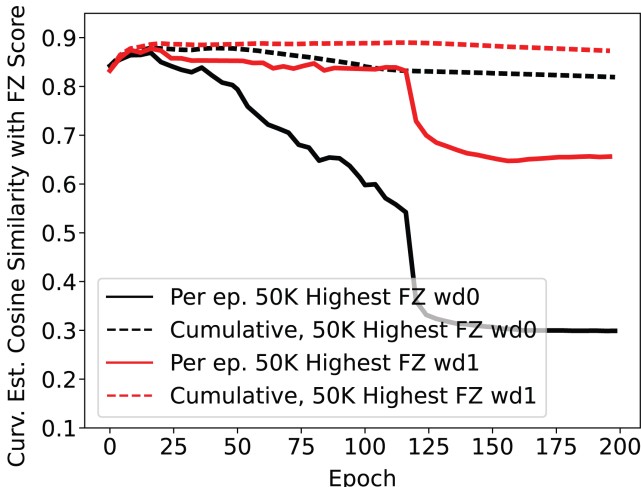

Figure 15: Cumulative and epoch-wise curvature of the top 50K FZ samples of ImageNet with and without weight decay.

Here we summarize the results of calculating curvature using the methodology from Garg & Roy (2023) with the main differences listed below

- Garg & Roy (2023) used adversarial direction to estimate curvature. We removed the adversarial direction assumption and returned to traditional Hutchinson's trace estimator form, and used random Rademacher vector instead for more reliable results.

- Garg & Roy (2023) calculated curvature at the end of training gives unreliable scores. For instance, the match with FZ scores for curvature calculated at the end of training is shown in Table 4, which is very low. We average curvature during training to get reliable results,

and to allow for different decision boundaries that have been learned at different epochs as
training progresses.

| Method | Top 5K FZ score cosine similarity | | Cosine similarity with FZ score for all data | |
|---|---|---|---|---|
| | wd0 | wd1 | wd0 | wd1 |
| Garg & Roy (2023) | 0.07 | 0.24 | 0.10 | 0.17 |
| Ours $n = 10$ @ End of Training | 0.06 | 0.28 | 0.12 | 0.18 |
| Ours | 0.82 | 0.90 | 0.73 | 0.82 |

Table 4: Comparing Garg & Roy (2023) and our method for capturing memorization scores on CI-
FAR100 dataset, with (wd1) and without weight (wd0) decay.

## D.5 Results for Different architectures

| Architecture | Test Set Acc | Train Acc | Top-K CS | CS |
|---|---|---|---|---|
| ResNet18 | 75.09% | 99.99% | 0.90 | 0.82 |
| DenseNet121 | 76.09% | 99.98% | 0.83 | 0.74 |
| VGG13 BN | 69.81% | 99.98% | 0.87 | 0.75 |
| Mobilenet V2 | 66.21% | 99.97% | 0.90 | 0.77 |

Table 5: Cosine similarity between FZ scores and curvature scores on all (CS) and Top 5000 FZ
score samples (Top-K CS) on CIFAR100 dataset.

| | ResNet18 | VGG13 BN | DenseNet121 | Mobilenet V2 |
|---|---|---|---|---|
| ResNet18 | 1.00 | 0.89 | 0.88 | 0.92 |
| VGG13 BN | 0.89 | 1.00 | 0.83 | 0.88 |
| DenseNet121 | 0.88 | 0.83 | 1.00 | 0.85 |
| Mobilenet V2 | 0.92 | 0.88 | 0.85 | 1.00 |

Table 6: Cosine similarity between curvature scores of various architectures for top 5000 FZ score
samples (Top-K CS) on CIFAR100 dataset.

| | ResNet18 | VGG13 BN | DenseNet121 | Mobilenet V2 |
|---|---|---|---|---|
| ResNet18 | 1.00 | 0.83 | 0.83 | 0.87 |
| VGG13 BN | 0.83 | 1.00 | 0.76 | 0.83 |
| DenseNet121 | 0.83 | 0.76 | 1.00 | 0.78 |
| Mobilenet V2 | 0.87 | 0.83 | 0.78 | 1.00 |

Table 7: Cosine similarity between curvature scores of various architectures for all samples (CS) on
CIFAR100 dataset.

## E Validation Samples with High Curvature

In this section, we train the same network as described in section 4 on the validation sets of all 4
datasets and show the highest curvature samples for these. Results are of curvature averaged over
all training epochs, and the network is trained without weight decay.

## F CIFAR100 most memorized samples

Here we show the hundred most memorized examples as identified by FZ scores, and with curvature
when training with weight decay $= 1e - 4$. Despite FZ score being 3 orders of magnitude more

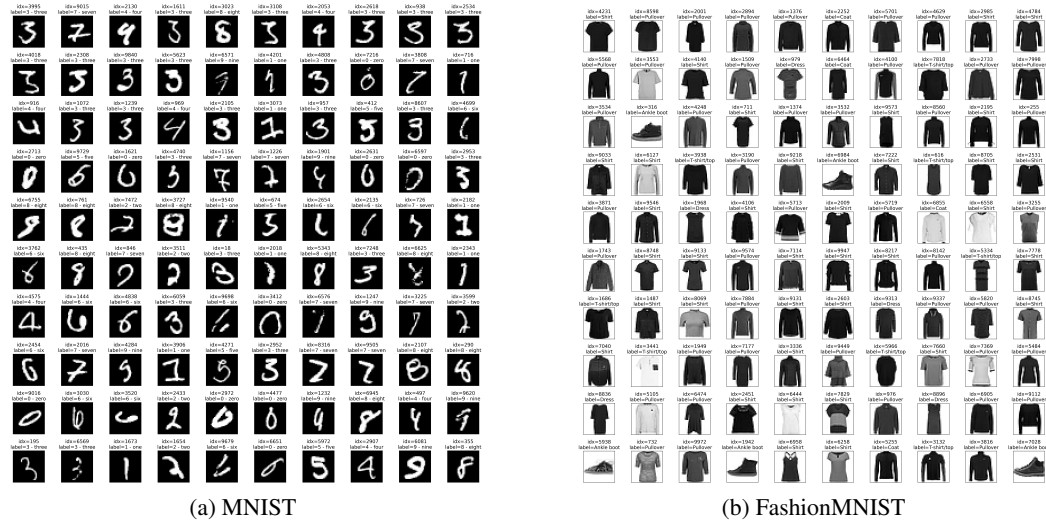

| (a) MNIST | (b) FashionMNIST |

Figure 16: High curvature samples from validation sets

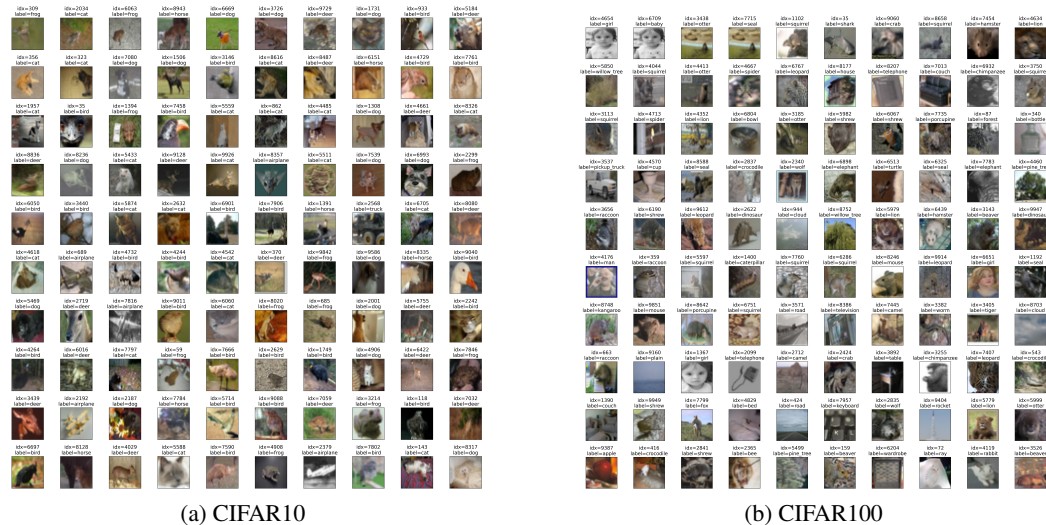

| (a) CIFAR10 | (b) CIFAR100 |

Figure 17: High curvature samples from validation sets

computationally expensive than our method, they do not find the failure case of duplicated samples with differing labels that we found with our analysis. This is significant since the duplicate samples are most likely memorized by a model if the model gets 100% accuracy since they have differing labels.

## G    MNIST LABEL CORRUPTION

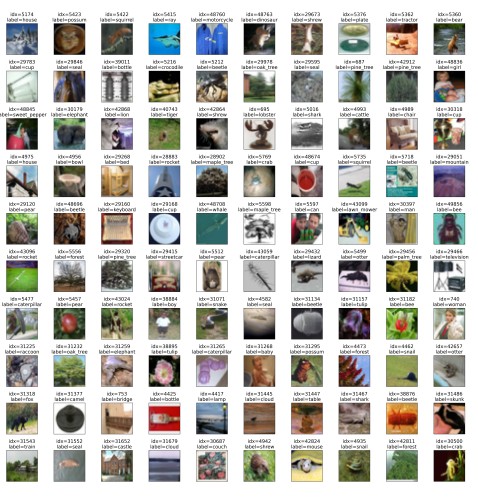

(a) Most Memorized according to FZ scores

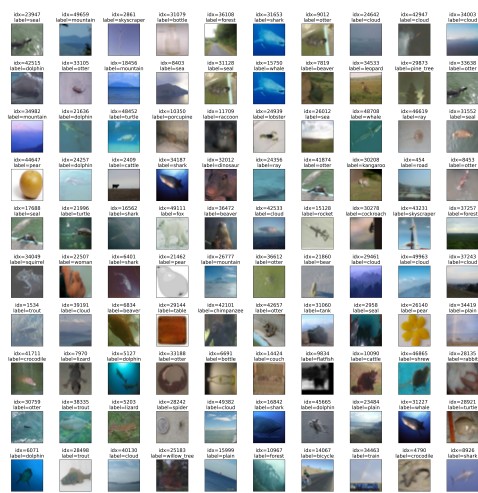

(b) Highest Curvature with Weight Decay on

Figure 18: High curvature samples from training sets of CIFAR100

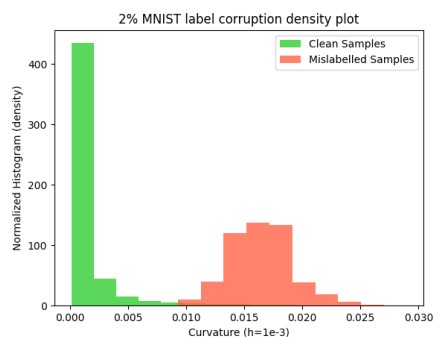

(a) 2% Label corruption, curvature density plot. Curvature obtained on modified ResNet18.

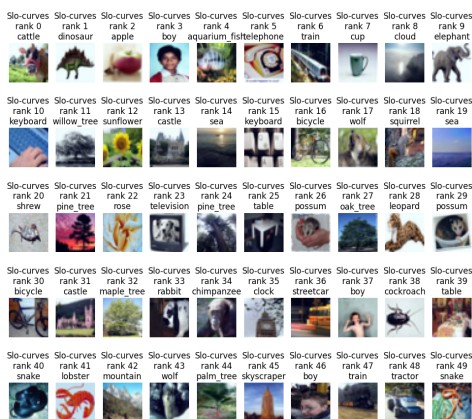

(b) High curvature samples from training set according to Slo-curves Garg & Roy (2023). Obtained of ResNet18 trained without weight decay on CIAR100

Figure 19: Visualizing MNIST label corruption histogram results using the proposed curvature averaging (left). High curvature samples according to Slo-curves Garg & Roy (2023) (Right)

