# OpenReview forum: "Memorization Through the Lens of Curvature of Loss Function Around Samples"
_ICLR.cc/2024/Conference — Submitted to ICLR 2024_

### Official Review · Reviewer_fwvr · 2023-10-18

**Soundness:** 2 fair
**Presentation:** 2 fair
**Contribution:** 2 fair
**Rating:** 5
**Confidence:** 4

**Summary:**

- The paper proposes using the curvature of loss function around each training sample, averaged over training epochs, as a scalable method, to measure of memorization of a given sample.
- The metric is validated both in the setting of synthetic label noise, as well as against an independent and
comprehensively calculated baseline of memorization scores released by Feldman & Zhang (2020)

**Strengths:**

- Reduces the time to calculate memorization scores considerably as compared to the baseline of Feldman & Zhang(though a head to head comparison doesn't make sense as they can also calculate influence scores in their technique).
- Decently high cosine similarity with FZ baseline and high AUROC values for identifying the noisy label samples.

**Weaknesses:**

- Scalability: To capture the curvature of training samples in every epoch, additional backward passes are required(n=10 for these datasets), which requires an additional 10X the training time to capture the metric proposed. It can be partially circumvented by calculating the curvature per sample every few epochs, but can still continue to be computationally expensive.
- Limited to memorization: The baseline by Feldman & Zhang calculates both the memorization scores for the training examples, and influence scores for test examples. There is no natural extension of curvature as a metric to influence scores.
- Baseline: No comparison with any baseline is presented. Simple baselines like learning time, second split forgetting time etc. should be compared to.
- Hyperparameters : No discussion related to how the values of h and n affect the results obtained

**Questions:**

- Results of comparison with other simple baselines ?

- How does the value of h and n in the curvature calculation affect the results ?

---

> ### Author Response · Authors · 2023-11-23
> **Response to Reviewer: Part 1/2**
>
> We thank the reviewer for their time and consideration, and answer the questions here.
> 1. Scalability: To capture the curvature of training samples in every epoch, additional backward passes are required(n=10 for these datasets), which requires an additional 10X the training time to capture the metric proposed. It can be partially circumvented by calculating the curvature per sample every few epochs, but can still continue to be computationally expensive.
>
> Authors: While it is true that we need upto n=10 extra forward and backward passes to calculate curvature as shown in the paper, it is still far more efficient than the baseline for calculating curvature scores, Feldman&Zhang, that requires training upto 20,000 models (equivalent to n=20,000 for us). Additionally, we show through our hyperparameter runs in the next answer, that we can use a smaller n (upto n=5) to get reliable results. We also show from our ImageNet model, that we can calculate curvature every 4 epochs, resulting in the extra computation reducing to just about 1x (double the standard runtime). Further, we perform analysis in Section 4 to show the earlier epochs give a much higher match with cosine similarity score, and resource constrained users can use this to determine the best subset of epochs to perform curvature calculations on. We believe this allows us to scale our method, as evidenced by our ability to show results for ImageNet dataset without using massive GPUs.
>
> &nbsp;
>
> 2. Limited to memorization: The baseline by Feldman & Zhang calculates both the memorization scores for the training examples, and influence scores for test examples. There is no natural extension of curvature as a metric to influence scores.
>
> Authors: We thank the reviewer for helping us think in this direction. We feel that using contrastive loss and calculating curvature on this loss might help us get influence scores. We leave this exploration for future work, but thank the reviewer for identifying another application of our method. We want to highlight that if we were to use contrastive loss, most methods mentioned in the prior work section would not be applicable (SSFT [Maini1], CL [NorthCutt1], Inconfidence [Carlini1], FZ [Feldman2], AUM [Pleiss1], Learning Time) since they require logits to work. However, since curvature only requires a loss, we would still be able to capture the scores.
>
> &nbsp;
>
> Feldman2 - Vitaly Feldman and Chiyuan Zhang. What neural networks memorize and why: Discovering the long tail via influence estimation. Advances in Neural Information Processing Systems, 33:2881–2891, 2020.
>
> Maini1 - Pratyush Maini, Saurabh Garg, Zachary Lipton, and J Zico Kolter. Characterizing datapoints via second-split forgetting. Advances in Neural Information Processing Systems, 35:30044–30057, 2022.
>
> NorthCutt1 - Curtis Northcutt, Lu Jiang, and Isaac Chuang. Confident learning: Estimating uncertainty in dataset labels. Journal of Artificial Intelligence Research, 70:1373–1411, 2021a.
>
> Carlini1 - Nicholas Carlini, Ulfar Erlingsson, and Nicolas Papernot. Distribution density, tails, and outliers in machine learning: Metrics and applications. arXiv preprint arXiv:1910.13427, 2019a.
>
> Pleiss1 - Geoff Pleiss, Tianyi Zhang, Ethan Elenberg, and Kilian Q Weinberger. Identifying mislabeled data using the area under the margin ranking. Advances in Neural Information Processing Systems, 33:17044–17056, 2020

---

> > ### Author Response · Authors · 2023-11-23
> > **Continued Response 2/2**
> >
> > 3. Baseline: No comparison with any baseline is presented. Simple baselines like learning time, second split forgetting time etc. should be compared to. and Question 1: Results of comparison with other simple baselines?
> >
> > Authors: We have added both these comparisons to Table 2. From Table 2, we can see that Curvature outperforms SSFT for both MNIST and CIFAR-10 for all percentages of mislabeling considered. However, SSFT performs better than Curvature for CIFAR-100, but as the authors note in their text, SSFT can be combined with other methodologies. We show that combining SSFT and Curvature results in a combination that outperforms both SSFT and Curvature independently and significantly. We have provided the same table below for reference, with the experiments for LT and SSFT performed using the author's github repo. Additionally, we also note that SSFT results shown in the paper fail to capture the novel failure mode we caught, that of duplicate samples with differing labels.
> >
> > &nbsp;
> > We thank the reviewer for pointing us to a strong baseline, and for helping us strengthen out paper by improving our results by combining it with SSFT. However, we would still like to reemphasize that our paper focuses on establishing curvature as a method for measuring memorization, and the application of detecting mislabeled samples is shown merely as corroboration that curvature does indeed detect memorization. There are not many baselines available for memorization, but we compare our method against the most established method currently, FZ scores, and show that we capture failure modes that they fail to, despite being 3 orders of magnitude more efficient.
> >
> > |  Dataset |   Method  | Corruption |         |         |        |        |        |
> > |:--------:|:---------:|:----------:|---------|---------|--------|--------|--------|
> > |          |           | 1%         | 2%      | 4%      | 6%     | 8%     | 10%    |
> > | MNSIT    | Inconf.   | 99.4\%     | 99.0\%  | 98.4\%  | 97.7\% | 97.1\% | 96.1\% |
> > |          | CL        | 99.7\%     | 99.3\%  | 99.1\%  | 98.9\% | 98.9\% | 98.9\% |
> > |          | SSFT      | 99.9\%     | 99.9\%  | 99.8\%  | 99.7\% | 99.7\% | 99.5\% |
> > |          | LT        | 98.7\%     | 99.5\%  | 98.4\%  | 98.5\% | 97.7\% | 97.3\% |
> > |          | Curv      | 100.0\%    | 100.0\% | 99.9\%  | 99.9\% | 99.9\% | 99.9\% |
> > | CIFAR10  | Inconf.   | 84.2\%     | 82.5\%  | 81.8\%  | 81.6\% | 81.5\% | 81.5\% |
> > |          | CL        | 92.4\%     | 93.1\%  | 93.4\%  | 93.4\% | 91.7\% | 93.4\% |
> > |          | SSFT      | 94.5\%     | 94.1\%  | 93.2\%  | 92.5\% | 91.6\% | 90.0\% |
> > |          | LT        | 87.3\%     | 82.5\%  | 84.0\%  | 83.4\% | 82.2\% | 83.0\% |
> > |          | Curv      | 97.4\%     | 96.6\%  | 95.5\%  | 94.4\% | 94.1\% | 92.9\% |
> > |          | Curv SSFT | 97.5\%     | 96.8\%  | 96.2\%  | 96.1\% | 95.2\% | 94.8\% |
> > | CIFAR100 | Inconf.   | 85.0\%     | 84.0\%  | 83.5\%  | 83.6\% | 83.6\% | 83.4\% |
> > |          | CL        | 83.1\%     | 84.2\%  | 85.3\%  | 86.3\% | 84.3\% | 84.6\% |
> > |          | SSFT      | 94.8\%     | 93.7\%  | 93.2\%  | 91.7\% | 91.9\% | 91.0\% |
> > |          | LT        | 85.2\%     | 85.2\%  | 81.8\%  | 80.8\% | 79.8\% | 78.5\% |
> > |          | Curv      | 90.9\%     | 89.6\%  | 88.3\%  | 86.8\% | 85.7\% | 84.3\% |
> > |          | Curv SSFT | 96.0\%     | 94.5\%  | 94.2\%  | 93.1\% | 92.9\% | 91.9\% |
> >
> >
> > &nbsp;
> > 4. Hyperparameters : No discussion related to how the values of h and n affect the results obtained. and Question: How does the value of h and n in the curvature calculation affect the results?
> >
> > Authors: We have added the hyperparameter tuning results in the appendix and provide the same in the table below. The hyperparameter tuning results show the cosine similarity match between FZ scores and curvature score for all (CS) and top 5000 (Top-K CS) examples on CIFAR10 dataset. From the Table we have the following take away, the proposed method is pretty robust to hyper parameter choices. Next, we see that the choice of n is less important than the choice of h. But overall, the scores are very robust.
> >
> > |       h      |      n  |      Top-K CS  |      CS   |
> > |--------------|---------|----------------|-----------|
> > |      0.01    |     5   |     0.89       |     0.80  |
> > |      0.001   |     5   |     0.90       |     0.82  |
> > |      0.0001  |     5   |     0.90       |     0.82  |
> > |      0.01    |     10  |     0.89       |     0.80  |
> > |      0.001   |     10  |     0.90       |     0.82  |
> > |      0.0001  |     10  |     0.90       |     0.82  |
> > |      0.01    |     20  |     0.89       |     0.80  |
> > |      0.001   |     20  |     0.90       |     0.82  |
> > |      0.0001  |     20  |     0.90       |     0.82  |

---

### Official Review · Reviewer_a9Na · 2023-10-19

**Soundness:** 3 good
**Presentation:** 3 good
**Contribution:** 2 fair
**Rating:** 5
**Confidence:** 4

**Summary:**

This paper propose to use a curvature inspired metric (trace of the squared Hessian of the per-example loss with respect to a particular input) to identify memorized examples. The paper show that this metric is consistent with a previous memorization metric (FZ scores) while being less computationally expensive. Moreover, the proposed metric, when trained without weight decay, identify a failure pattern of near identical images with different labels that is not extensively studied before. The metric is also shown to outperform several baselines in corrupted label detection and provide insights to learning dynamics at the presence of label noises from the perspective of curvatures.

**Strengths:**

This paper propose a simple curvature inspired metric with an approximation procedure that can be computed efficiently. This metric is shown to be effective at identifying examples memorized by neural networks. The idea is simple and the presentation is easy to follow. The experiment setup are clear and the results are sound.

**Weaknesses:**

1. There are a large body of work on data valuation for deep neural networks that studies similar questions. Those should be discussed in the related work and at least representative methods should be compared in the experiment.

2. This paper could benefit from a stronger demonstration of the utility of the proposed metric, beyond the simple corrupted label prediction experiment. A few hypothetical directions could be

    1. Could the curvature be extended to compute some kind of influence metrics from training to test examples like the FZ scores or other data valuation metrics?

    2. Theoretical analysis of how or why is the proposed metric connected to memorization and generalization.

    3. New algorithms designs, such as curriculum learning based on per-example curvatures, or maybe regularizers motivated by the curvature metric.

3. Unlike other metrics that measures in the output, logits, or weight space, the proposed metric probably depend more on the geometry of the inputs. Therefore, I think it is very valuable if the paper could present studies with different input domains.

**Questions:**

1. Do you observe the same behavior if the input-vs-logit space analysis in Fig. 7 is performed on a real dataset instead of a synthetic one?

2. It is described in the appendix how the two hyperparameters h and n are tuned. Can you show some hyperparameter tuning curves? In particular, I'm interested in not only what values are chosen, but also how robust the proposed metric is to hyperparameter changes.

---

> ### Author Response · Authors · 2023-11-23
> **Response Part 1/3: Generic Response**
>
> ### Response to Reviewer: Part 1/2
> We thank the reviewer for their time and consideration. We would like to address the overall review here, and then go into the details of each question after that.
> &nbsp;
>
> We want to emphasize that in this paper we wanted to introduce the idea of measuring memorization with input loss curvature. We believe that extreme memorization and overfitting is one of the most significant insights into deep neural networks, as evidenced by the extensive work done in references [Zhang1, Feldman1, Feldman2].  We establish curvature as a reliable and efficient metric to measure memorization, and show both quantitative and qualitative corroborations of the reliability of curvature as a memorization metric.
> &nbsp;
>
>
> For practical applications, we show three in the paper which we believe are quite significant. One is detecting mislabeled samples, and that is an area of significant research [Maini1, Northcutt1, Carlini1]. Secondly, we show an as of yet unobserved failure mode on CIFAR100 and ImageNet, two of the most widely used datasets to test different algorithms for vision algorithms, that of duplicate samples with differing labels. This shows how curvature can be used to find failure modes especially for datasets used in the industry that may not have been vetted by many research works. Thirdly, we also note the outliers in FashionMNIST dataset show a bias towards high contrast clothes, despite them being not more prototypical of clothing in real life. This highlights that our analysis can reveal biases present in the dataset.
> &nbsp;
>
>
> There are many downstream applications of being able to measure and study memorization, and the reviewer lists some that we are currently working on as extensions. However, this work forms the bases of all those future explorations, which we and hopefully the community will pursue. We feel that this manuscript is self-contained in its claim, justification and example applications, and hope that the reviewer would reconsider the paper on its own merit, instead of the potential applications that can exist as a result of this manuscript.
> &nbsp;
>
> References:
> Zhang1 - Chiyuan Zhang, Samy Bengio, Moritz Hardt, Benjamin Recht, and Oriol Vinyals. “Understanding deep learning requires rethinking generalization”. In ICLR 2017,
>
> Feldman1 - Vitaly Feldman. Does learning require memorization? a short tale about a long tail. In Proceedings of the 52nd Annual ACM SIGACT Symposium on Theory of Computing, pp. 954–959, 2020.
>
> Feldman2 - Vitaly Feldman and Chiyuan Zhang. What neural networks memorize and why: Discovering the long tail via influence estimation. Advances in Neural Information Processing Systems, 33:2881–2891, 2020.
>
> Maini1 - Pratyush Maini, Saurabh Garg, Zachary Lipton, and J Zico Kolter. Characterizing datapoints via second-split forgetting. Advances in Neural Information Processing Systems, 35:30044–30057, 2022.
>
> NorthCutt1 - Curtis Northcutt, Lu Jiang, and Isaac Chuang. Confident learning: Estimating uncertainty in dataset labels. Journal of Artificial Intelligence Research, 70:1373–1411, 2021a.
>
> Carlini1 - Nicholas Carlini, Ulfar Erlingsson, and Nicolas Papernot. Distribution density, tails, and outliers in machine learning: Metrics and applications. arXiv preprint arXiv:1910.13427, 2019a.
> &nbsp;
> &nbsp;
> &nbsp;

---

> > ### Author Response · Authors · 2023-11-23
> > **Continued Response 2/3: Specific Questions**
> >
> > ### Questions and Responses
> > 1. There are a large body of work on data valuation for deep neural networks that studies similar questions. Those should be discussed in the related work and at least representative methods should be compared in the experiment.
> >
> > Authors: We have added comparisons to two new baselines, Learning Time (LT) and SSFT [Maini1]  in Table 2. From Table 2, we can see that Curvature outperforms LT for all datasets, and  SSFT for both MNIST and CIFAR-10 for all percentages of mislabeling considered. However, SSFT performs better than Curvature for CIFAR-100, but as the authors note in their text, SSFT can be combined with other methodologies. We show that combining SSFT and Curvature results in a combination that outperforms both SSFT and Curvature independently and significantly. Additionally, we also note that SSFT results shown in the paper fail to capture the novel failure mode we caught, that of duplicate samples with difffering labels.
> >
> > &nbsp;
> >
> > We thank the reviewer for encouraging us to pursue new baselines and helping us strengthen our paper by improving our results by combining it with SSFT.
> > &nbsp;
> >
> > Maini1 - Pratyush Maini, Saurabh Garg, Zachary Lipton, and J Zico Kolter. Characterizing datapoints via second-split forgetting. Advances in Neural Information Processing Systems, 35:30044–30057, 2022.
> > ### Regarding Potential Extensions:
> >
> > 1. Could the curvature be extended to compute some kind of influence metrics from training to test examples like the FZ scores or other data valuation metrics?
> >
> >  Authors:  We thank the reviewer for helping us think in this direction. We feel that using contrastive loss and calculating curvature on this loss might help us get influence scores. We leave this exploration for future work, but thank the reviewer for identifying another application of our method. We want to highlight that if we were to use contrastive loss, most methods mentioned in the prior work section would not be applicable (SSFT [Maini1], CL [NorthCutt1], Inconfidence [Carlini1], FZ [Feldman2], AUM [Pleiss1], Learning Time) since they require logits to work. However, since curvature only requires a loss, we would still be able to capture the scores.
> >
> >
> > Feldman2 - Vitaly Feldman and Chiyuan Zhang. What neural networks memorize and why: Discovering the long tail via influence estimation. Advances in Neural Information Processing Systems, 33:2881–2891, 2020.
> >
> > Maini1 - Pratyush Maini, Saurabh Garg, Zachary Lipton, and J Zico Kolter. Characterizing datapoints via second-split forgetting. Advances in Neural Information Processing Systems, 35:30044–30057, 2022.
> >
> > NorthCutt1 - Curtis Northcutt, Lu Jiang, and Isaac Chuang. Confident learning: Estimating uncertainty in dataset labels. Journal of Artificial Intelligence Research, 70:1373–1411, 2021a.
> >
> > Carlini1 - Nicholas Carlini, Ulfar Erlingsson, and Nicolas Papernot. Distribution density, tails, and outliers in machine learning: Metrics and applications. arXiv preprint arXiv:1910.13427, 2019a.
> >
> > Pleiss1 - Geoff Pleiss, Tianyi Zhang, Ethan Elenberg, and Kilian Q Weinberger. Identifying mislabeled data using the area under the margin ranking. Advances in Neural Information Processing Systems, 33:17044–17056, 2020
> >
> > &nbsp;
> >
> > 2. Theoretical analysis of how or why is the proposed metric connected to memorization and generalization.
> >
> > Authors: We are working on establishing theoretical results in a follow up work. However, we feel the claims and results in this paper stand on their own.
> >
> > &nbsp;
> >
> > 3. New algorithms designs, such as curriculum learning based on per-example curvatures, or maybe regularizers motivated by the curvature metric.
> >
> > Authors: This is a very interesting idea, with wide applications. We are in the process of exploring some of the applications. That being said, we establish in this paper that studying curvature is useful and hope the community can explore different applications.
> >
> > &nbsp;
> >
> > 4. Unlike other metrics that measures in the output, logits, or weight space, the proposed metric probably depend more on the geometry of the inputs. Therefore, I think it is very valuable if the paper could present studies with different input domains.
> >
> > Authors: We agree with the reviewer. We think this discussion highlights the advantages of our method, namely it only requires being able to calculate loss, instead of logits which may not always be available. Thus, it can be extended to different domains and application with relative ease. We hope it will encourage the community to explore input loss curvature for different domains.
> >
> > &nbsp;

---

> > > ### Author Response · Authors · 2023-11-23
> > > **Continued Response 3/3**
> > >
> > > 5. Do you observe the same behavior if the input-vs-logit space analysis in Fig. 7 is performed on a real dataset instead of a synthetic one?
> > >
> > >  Authors: Visualizing the input vs logit space is tricky for real datasets due to the higher dimensional nature of both the input data (3072 dimensional for CIFAR100) and of logits (100 for CIFAR100). However, we see the same curvature trend for our toy dataset and CIFAR100 and ImageNet datasets. As a proxy for logit space one can look at softmax confidence which is used in [NorthCutt1], and learning time. These studies tend to suggest that logit margin maximization tends to happen on real datasets as well. However, it would be hard to visualize as clearly as in our toy dataset.
> > >
> > > &nbsp;
> > >
> > > NorthCutt1 - Curtis Northcutt, Lu Jiang, and Isaac Chuang. Confident learning: Estimating uncertainty in dataset labels. Journal of Artificial Intelligence Research, 70:1373–1411, 2021a.
> > >
> > > &nbsp;
> > >
> > >
> > > 6. It is described in the appendix how the two hyperparameters h and n are tuned. Can you show some hyperparameter tuning curves? In particular, I'm interested in not only what values are chosen, but also how robust the proposed metric is to hyperparameter changes.
> > >
> > >  Authors: We have added the hyperparameter tuning results in the appendix and provide the same in the table below. The hyperparameter tuning results show the cosine similarity match between FZ scores and curvature score for all (CS) and top 5000 (Top-K CS) examples on CIFAR10 dataset. From the Table we have the following take away, the proposed method is pretty robust to hyper parameter choices. Next, we see that the choice of n is less important than the choice of h, and would recommend not using an h of less than 0.001.
> > > &nbsp;
> > >
> > > |       h      |      n  |      Top-K CS  |      CS   |
> > > |--------------|---------|----------------|-----------|
> > > |      0.01    |     5   |     0.89       |     0.80  |
> > > |      0.001   |     5   |     0.90       |     0.82  |
> > > |      0.0001  |     5   |     0.90       |     0.82  |
> > > |      0.01    |     10  |     0.89       |     0.80  |
> > > |      0.001   |     10  |     0.90       |     0.82  |
> > > |      0.0001  |     10  |     0.90       |     0.82  |
> > > |      0.01    |     20  |     0.89       |     0.80  |
> > > |      0.001   |     20  |     0.90       |     0.82  |
> > > |      0.0001  |     20  |     0.90       |     0.82  |

---

> > > > ### Comment · Reviewer_a9Na · 2023-12-04
> > > > **Thanks for the response**
> > > >
> > > > I would like to thank the authors for their detailed response and discussions. I'm glad that the authors found some of the directions valuable for future exploration. I believe the paper could be more exciting of some of those future explorations could be incorporated. However, for the current version of the paper, I would like to keep my original rating.

---

### Official Review · Reviewer_1WGU · 2023-10-31

**Soundness:** 2 fair
**Presentation:** 2 fair
**Contribution:** 2 fair
**Rating:** 5
**Confidence:** 4

**Summary:**

This paper studies memorization in modern neural networks and aims to design a tractable approach to quantify the memorization of different training examples. Specifically, this paper considers loss curvature (averaged across different training epochs) with respect to a given training example as a memorization measure for that particular example. The paper leverages a finite step approximation of the per-example loss Hessian and explores the proposed curvature-based memorization score on MNIST, Fashion-MNIST, CIFAR-10/100, and ImageNet, with ResNet18 as the underlying model architecture. Visualizing the examples with high curvature-based memorization scores, the authors observed that such examples correspond to long-tailed, mislabeled, or conflicting examples (e.g., duplicate examples with different labels). The paper also compared the curvature-based memorization score with Feldman and Zhang’s stability-based memorization score and found a high correlation between the two scores. For a synthetic setup, where label noise is introduced to a fraction of training examples, the paper shows that noisy examples belong to the set of examples that receive the highest curvature-based memorization scores. Finally, the paper studies the average (over the entire dataset) curvature of per-example losses as training evolves. The paper shows that, during the overfitting phase of training, this average curvature shows a decreasing trend while the validation loss keeps increasing. The paper provides an explanation for this seemingly contradictory observation by arguing the decrease in the average curvature corresponds to the phase where training examples are already classified correctly and the model is increasing the margin of these examples. Thus, the paper highlights the importance of averaging the per-example loss curvature across entire training epochs to obtain an informative memorization score.

**Strengths:**

1) The paper studies an important problem of quantitatively characterizing the memorization behavior of various training examples. The paper proposes the per-example loss curvature (averaged across different training epochs) as such a quantitative measure.
2) The paper successfully shows that the proposed memorization score can consistently identify a range of overfitting (“memorization”)-prone examples across multiple image classification benchmarks. This suggests that the proposed score can be utilized for cleaning/denoising a given training dataset.
3) The computational cost of evaluating the proposed memorization score is smaller than some of the other widely accepted memorization scores in the literature, such as Feldman & Zhang’s score.
4) The paper carefully studies the behavior of per-example loss curvature as training progresses and highlights the importance of averaging the per-example loss curvature across multiple training epochs.

**Weaknesses:**

1) The main weakness of the paper is that it does not provide a comprehensive discussion on what it means to say that an example is being memorized. For example, the work of Feldman & Zhang calls those points to be memorized where the model only performs well when those points are present in the training set. What is the precise notion of memorization that this paper aims to capture?
2) It appears that the paper claims that their curvature-based memorization score aligns well with Feldman & Zhang’s memorization score as the two scores have a high correlation (cosine similarity). Such aggregate-level correlations can be misleading, especially while studying memorization behaviors which are very much tied to individual examples (e.g., see Section 3.5 in https://arxiv.org/pdf/2310.05337.pdf).
3) Most of the experiments in the paper are tied to a single model. It would be interesting to see how the proposed curvature-based memorization score behaves as one changes model architecture and/or size. Are key takeaways from this paper robust to such variations?
4) The paper claims that their proposed memorization score is significantly cheaper to compute compared to existing scores like Feldman & Zhang’s memorization score (see a question on this below). However, it appears that the *efficient* loss curvature calculation follows from the prior work, limiting the technical novelty of the contributions.
5) There is significant scope for improvement in the quality of the presentation. For instance

* In Section 3.1, both $d$ and $D$ are used to represent the input dimension.
* In the line after Eq. (2) $v_i$ represents the $i$-th coordinate of the random Rademacher vector $v$, while in Eq (3), $v_i$ represents $i$ random *vector*. Also, the line after Eq. (3) refers to $v$ as a Rademacher *variable* instead of *vector* (a similar issue in Section 3.2).
* In Eq. (4), please consider using $\approx$ instead of $=$.
* In Eq. (6), please make the dependence of the square norm on indices $i$ (via $v_i$) and $t$ (via $W_t$) explicit.
* In Section 3.2, O(n) → O(nT) forward and backward passes?

Besides the issues mentioned above, multiple sentences in the paper require paraphrasing as their meaning is not entirely clear, and thorough proofreading is required to eliminate various typos.

**Questions:**

1) How does a memorizing network behave on duplicate examples with distinct labels? Which label is preferred by the model? Does the model encounter a decreasing loss curvature trend on such examples as training progresses?
2) In Section 4.2, the authors state ``... These scores are likely to be independent of spurious correlations to curvature that other methods such as confidence of prediction might have, and hence serve as a good baseline.`` Could the authors elaborate on this statement on why Feldman & Zhang’s score is likely to be independent of spurious correlations (while other methods can potentially exhibit such correlations)?
3) In Section 4.2, the authors claim that computing Feldman & Zhang (FZ) score is ``~3 x more computationally expensive``. Why is it **only** 3x more expensive when computing FZ scores requires training a large number of models?
4) In Section 4.3, the authors mention ``...We recommend that curvature analysis should be used in conjunction with other checks, for a holistic view of dataset integrity. ``. Could the authors clarify which other checks they refer to?

---

> ### Author Response · Authors · 2023-11-23
> **Response to Reviewer: Part 1/4**
>
> We thank the reviewer for taking the time to write detailed feedback. We answer the questions here:
> &nbsp;
>
> 1. The main weakness of the paper is that it does not provide a comprehensive discussion on what it means to say that an example is being memorized. For example, the work of Feldman & Zhang calls those points to be memorized where the model only performs well when those points are present in the training set. What is the precise notion of memorization that this paper aims to capture?
>
> Authors: This is a very relevant question, and we attempt to share the way we think of our metric with the reviewer here. The memorization that our manuscript is aiming to capture is conceptually quite similar to what Feldman and Zhang intend to capture with their metric. Specifically, both methods ask how sensitive the model is to a particular input. Feldman & Zhang propose to answer this by looking at probabilities in presence and absence of the sample, and we propose to answer this by measuring the sensitivity of the model to perturbation in the input, calculated via curvature.
> &nbsp;
>
>
>  A far more intuitive understanding of why we chose curvature arises from correlating the curvature of loss around a sample to the curvature of decision boundary around the sample in the input space. The classical machine learning illustration of overfitting, such as the one found [here](https://media.geeksforgeeks.org/wp-content/cdn-uploads/20190523171258/overfitting_2.png), shows that the network overfits by bending the decision boundary around certain sample points in the input space, indicating that these samples are being memorized by the network. In the example shown, the samples that would be counted as most memorized, would be the ones with the curviest decision boundaries around them, establishing a straightforward link between memorization score and curvature of decision boundary around the sample.
> &nbsp;
>
>
> Previous work such as [Dong1, Srinivas1] has shown the link between curvature of decision boundary and curvature of loss, in that regularizing loss curvature results in a flatter decision boundary. This link means that our metric estimates the curvature of the decision boundary around a sample in the input space, which makes intuitive sense as a measure of memorization.
> &nbsp;
>
>
>
> If the reviewer feel  that this would add clarity and intuitive visualization of our metric to the readers, we are happy to add this to the manuscript.
>
> &nbsp;
> References:
> Dong1 - Bin Dong, Haocheng Ju, Yiping Lu, and Zuoqiang Shi. "CURE: Curvature regularization for missing data recovery." SIAM Journal on Imaging Sciences 13, no. 4 (2020): 2169-2188.
>
> Srinivas1 Suraj Srinivas, Kyle Matoba, Himabindu Lakkaraju, and Francois Fleuret. "Efficient training of low-curvature neural networks." Advances in Neural Information Processing Systems 35 (2022): 25951-25964.
>
> &nbsp;
>
> 2. It appears that the paper claims that their curvature-based memorization score aligns well with Feldman & Zhang’s memorization score as the two scores have a high correlation (cosine similarity). Such aggregate-level correlations can be misleading, especially while studying memorization behaviors which are very much tied to individual examples (e.g., see Section 3.5 in [https://arxiv.org/pdf/2310.05337.pdf](https://arxiv.org/pdf/2310.05337.pdf)).
>
> Authors: We agree with the reviewer, and this is a justified question. In order to inform the readers, we have added a qualifying line to state the limitations of cosine similarity in the paper. However, we list two points in support of why we used cosine similarity below:
> * First, we show in Table 4 in the Appendix D, curvature scores calculated only at the end of training do not show a high cosine similarity match with FZ scores, with the cosine similarity being only 0.18 compared to our match of 0.82 when averaged over all epochs. We know that the curvature scores calculated at the end of training do not capture the duplicated samples, and are therefore not reliable. The fact that we get visually more reliable results by averaging over epochs, and our cosine similarity match rises significantly (from 0.18 to 0.82), leads us to believe that the cosine similarity metric still captures meaningful correlations.
> * Additionally, we show an independent link between FZ scores and curvature that does not rely on cosine similarity in Figure 8: We plot the curvature of the 5000 most memorized samples as per FZ scores and show that the curvature of FZ score identified memorized samples is considerably higher at all epochs.
> &nbsp;
> These two reasons were why we decided to proceed with the cosine similarity score in the paper.

---

> > ### Author Response · Authors · 2023-11-23
> > **Continued Response 2/4**
> >
> > 3. Most of the experiments in the paper are tied to a single model. It would be interesting to see how the proposed curvature-based memorization score behaves as one changes model architecture and/or size. Are key takeaways from this paper robust to such variations?
> >
> >     Authors: In addition to ResNet18 we provide results on VGG 13 (with Batch Norm), DenseNet121 and MobileNetV2 for CIFAR100, trained with a weight decay of 1e-4. Curvature scores from each network is able to obtain a good match with FZ scores.  We also provide cosine similarity match cross architecture (between arch1 and arch2) for all the samples and for top 5000 FZ score samples in CIFAR100 in the following tables. These results have also been added to the appendix of the paper. The results suggest that curvature is quite robust, since we observe a good match between FZ and curvature score across different architecture and model sizes, including architectures withlower test set accuracy.
> >
> >     &nbsp;
> >
> >     |       Architecture  |      Test Set Acc  |      Train Acc  |      CS Top 5K FZ Score  |      CS   |
> >     |---------------------|--------------------|-----------------|--------------------------|-----------|
> >     |      ResNet18       |     75.09%         |     99.99%      |     0.90                 |     0.82  |
> >     |      DenseNet121    |     76.09%         |     99.98%      |     0.83                 |     0.74  |
> >     |      VGG13 BN       |     69.81%         |     99.98%      |     0.87                 |     0.75  |
> >     |      Mobilenet V2   |     66.21%         |     99.97%      |     0.90                 |     0.77  |
> >
> >     &nbsp;
> >
> >     Cosine similarity between curvature scores of various architectures for top 5000 FZ score samples (Top-K CS) on CIFAR100 dataset (below).
> >
> >     |                    |      ResNet18  |      VGG13 BN  |      DenseNet121  |      Mobilenet V2  |
> >     |--------------------|----------------|----------------|-------------------|--------------------|
> >     |      ResNet18      |     1.00       |     0.89       |     0.88          |     0.92           |
> >     |      VGG13 BN      |     0.89       |     1.00       |     0.83          |     0.88           |
> >     |      DenseNet121   |     0.88       |     0.83       |     1.00          |     0.85           |
> >     |      Mobilenet V2  |     0.92       |     0.88       |     0.85          |     1.00           |
> >
> >     &nbsp;
> >
> >     Cosine similarity between curvature scores of various architectures for all samples on CIFAR100 dataset (below).
> >     |                    |      ResNet18  |      VGG13 BN  |      DenseNet121  |      Mobilenet V2  |
> >     |--------------------|----------------|----------------|-------------------|--------------------|
> >     |      ResNet18      |     1.00       |     0.83       |     0.83          |     0.87           |
> >     |      VGG13 BN      |     0.83       |     1.00       |     0.76          |     0.83           |
> >     |      DenseNet121   |     0.83       |     0.76       |     1.00          |     0.78           |
> >     |      Mobilenet V2  |     0.87       |     0.83       |     0.78          |     1.00           |

---

> > > ### Author Response · Authors · 2023-11-23
> > > **Continued Response 3/4**
> > >
> > > 4. The paper claims that their proposed memorization score is significantly cheaper to compute compared to existing scores like Feldman & Zhang’s memorization score. However, it appears that the efficient loss curvature calculation follows from the prior work, limiting the technical novelty of the contributions.
> > >
> > >     Authors: Thank you for your question. Garg and Roy [R2] indeed was a source of inspiration for this work. We found their analysis of curvature for creating coresets interesting, and were inspired to see if we could use it as a measure of memorization. We found that their method did not work out of the box, and we had to make significant changes to establish a good metric, in essence reverting back to Hutchinson’s trace estimator form for calculating curvature [Hutchinson1]. Importantly, *their scores also failed to capture the failure mode of duplicated samples with differing labels*. We would like to highlight the following differences:
> > >
> > >     * In this manuscript, we focus on using curvature as a method for measuring memorization of  samples, and focus on the high curvature samples. In contrast, Garg and Roy 2023 [R2] focus on low curvature samples, and their application to data efficiency for coreset creation.
> > >
> > >     * We note that their method of calculating curvature does not give reliable results:
> > >         *   They used the adversarial direction to estimate curvature. We removed the adversarial direction assumption and returned to traditional Hutchinson’s trace estimator form, and used random Rademacher vectors instead for more reliable results. The cosine similarity with FZ scores on CIFAR100 for Garg and Roy [R2] is 0.17, while our method achieves 0.82.
> > >
> > >         * To further improve estimation, we average our results over 10 different random Rademacher variables while [R2] use a single direction. We get a good match for n=5,10,20 with FZ scores as seen in the hyperparameter tuning results from Table 3 in Appendix B.
> > >
> > >         * Their method only calculates curvature at the end of training, and the following table shows that that gives unreliable scores. For instance, the match with FZ scores for curvature calculated at the end of training, as shown in the table below, is only 0.18.  We average curvature during training to get reliable results, in order to allow for different decision boundaries that have been learned at different epochs as training progresses. This is the result of studying curvature dynamics during training in Section 4.4 and is quite evident if we form a GIF of the decision boundary in the input space. We have added that to the supplementary material, also found [here](https://imgur.com/a/IGDqwo0). The early epochs in the GIF show different hypotheses tried out by the network during the training process, and averaging over them allows us to not be over-reliant on any one hypothesis.
> > >
> > >         * We show the results for using Garg and Roy’s method, our method limited to calculating curvature at end of training, and our method as outlined in the paper and show that we get considerably better matches (better cosine similarity CS) with FZ scores for both the most memorized samples, and for the entire dataset, both with (wd1) and without (wd0) weight decay.
> > >             &nbsp;
> > >
> > >
> > >             |       Method            |      Top 5K FZ CS (wd0)      |   Top 5K FZ CS (wd1)        |      CS with FZ for all data (wd0) |     CS with FZ for all data (wd1)      |
> > >             |-------------------------|------------------------------|-----------------------------|------------------------------------|----------------------------------------|
> > >             |      Garg and Roy [R2]  |     0.07                     |     0.24                    |     0.10                           |     0.17                               |
> > >             |      Ours n=10 EOT      |     0.06                     |     0.28                    |     0.12                           |     0.18                               |
> > >             |      Ours               |     0.82                     |     0.90                    |     0.73                           |     0.82                               |
> > >
> > >         * We provide insights and strong empirical evidence for the use of high curvature samples in studying memorization of deep neural nets, and utilize it to identify a new failure mode that all other methods, including R2, fails to capture, as shown in Figure 19 in the Appendix.
> > >     &nbsp;
> > >
> > >     We thank the reviewer for helping us consolidate the differences, and have summarized this discussion in Appendix D4.

---

> > > > ### Author Response · Authors · 2023-11-23
> > > > **Continued Response 4/5**
> > > >
> > > > 5. There is significant scope for improvement in the quality of the presentation. For ... thorough proofreading is required to eliminate various typos.
> > > > Authors: Thank you we have updated the paper.
> > > > &nbsp;
> > > >
> > > > 6. How does a memorizing network behave on duplicate examples with distinct labels? Which label is preferred by the model? Does the model encounter a decreasing loss curvature trend on such examples as training progresses?
> > > >
> > > > Authors: We believe that this is hard to predict, and highly dataset and sample dependent.  For instance, let’s look at the case where the same sample is labeled girl and baby in CIFAR100. We think different models might choose different labels, but one major factor that would impact this choice would be the number of samples in each of the conflicting classes, with the majority class having a higher chance of being predicted. Another factor would be if there is another similar image with an embedding close to our conflicted sample, as that would influence the model to preferentially predict its class for the conflicting sample as well. Further, for the case in ImageNet where there is an image labeled both wine and corkscrew, since both are present in the image, we believe that the area occupied by each class might also play a role. Overall, we believe that this is a very interesting question, but would end up being largely dataset and sample dependent.
> > > >
> > > > &nbsp;
> > > >
> > > > 7. In Section 4.2, the authors state ... These scores are likely to be independent of spurious correlations to curvature that other methods such as confidence of prediction might have, and hence serve as a good baseline. Could the authors elaborate on this statement on why Feldman & Zhang’s score is likely to be independent of spurious correlations (while other methods can potentially exhibit such correlations)?
> > > >
> > > > Authors: We meant that for baseline comparison, we wanted to use a method that is unrelated to curvature in concept or calculation. In concept, the curvature likely has some correlation to distance from boundary, which other methods like the inconfidence value is also correlated to. For having a truly unrelated measure, we use Feldman and Zhang since their description of memorization is simple and depends on the prediction on a sample in presence and absence of it during training. It does not depend on training statistics and hence cannot be correlated to curvature or other related training metrics, and therefore serves as a largely independent baseline.
> > > >
> > > > &nbsp;
> > > >
> > > > 8. In Section 4.2, the authors claim that computing Feldman & Zhang (FZ) score is ~3 x more computationally expensive . Why is it only 3x more expensive when computing FZ scores requires training a large number of models?
> > > >
> > > > Authors: Thank you for identifying a major typo! We meant 3 orders of magnitude. We have fixed this in the paper. To clarify, our method complexity can be thought of as equivalent to training 10 standard models since we use n=10, while FZ method trains ~20000 models.
> > > >
> > > > &nbsp;
> > > >
> > > > 9. In Section 4.3, the authors mention ...We recommend that curvature analysis should be used in conjunction with other checks, for a holistic view of dataset integrity. . Could the authors clarify which other checks they refer to?
> > > >
> > > > Authors: We meant that there are many unknown failure modes that unvetted datasets can suffer from. Before we did the curvature analysis, we were not aware of the failure mode of duplicate samples with differing labels. There can be other failure modes that we do not anticipate that our method might fail to capture, and hence we recommend performing holistic checks. To begin with, we recommend performing the checks we use to compare our method’s performance against in Table 2, Confident Learning [NorthCutt1] and Inconfidence score [Carlini1]. Other scores can be found in [Pleiss1], [Katharopoulos1], [Johnson1] as mentioned in the prior work, and maybe more or less applicable depending on the task and data style. Additionally, we added two new baselines to Table 2 based on reviewer feedback, that of Learning Time and SSFT [Maini1] and show that our method combined with SSFT gives better results than both methods independently. As demonstrated, these combinations can help us find failure modes that individual methods might be insensitive to, and that might cause impactful problems in the decisions made by these models when they are deployed.

---

> > > > > ### Author Response · Authors · 2023-11-23
> > > > > **References**
> > > > >
> > > > > NorthCutt1 - Curtis Northcutt, Lu Jiang, and Isaac Chuang. Confident learning: Estimating uncertainty in dataset labels. Journal of Artificial Intelligence Research, 70:1373–1411, 2021a.
> > > > >
> > > > > Carlini1 - Nicholas Carlini, Ulfar Erlingsson, and Nicolas Papernot. Distribution density, tails, and outliers in machine learning: Metrics and applications. arXiv preprint arXiv:1910.13427, 2019a.
> > > > >
> > > > > Pleiss1 - Geoff Pleiss, Tianyi Zhang, Ethan Elenberg, and Kilian Q Weinberger. Identifying mislabeled data using the area under the margin ranking. Advances in Neural Information Processing Systems, 33:17044–17056, 2020
> > > > >
> > > > > Katharopoulos1 - Angelos Katharopoulos and Francois Fleuret. Not all samples are created equal: Deep learning with importance sampling. In International conference on machine learning, pp. 2525–2534. PMLR, 2018.
> > > > >
> > > > > Johnson1 - Tyler B Johnson and Carlos Guestrin. Training deep models faster with robust, approximate importance sampling. Advances in Neural Information Processing Systems, 31, 2018
> > > > >
> > > > > Maini1 - Pratyush Maini, Saurabh Garg, Zachary Lipton, and J Zico Kolter. Characterizing datapoints via second-split forgetting. Advances in Neural Information Processing Systems, 35:30044–30057, 2022.
> > > > >
> > > > > Hutchinson1 - Michael F Hutchinson. A stochastic estimator of the trace of the influence matrix for Laplacian smoothing splines. Communications in Statistics-Simulation and Computation, 19(2):433–450, 1990.
> > > > >
> > > > > R2 - Isha Garg and Kaushik Roy. Samples with low loss curvature improve data efficiency. In Proceedings of the IEEE/CVF Conference on Computer Vision and Pattern Recognition, pp. 20290–20300, 2023.

---

> > > > > > ### Comment · Reviewer_1WGU · 2023-12-03
> > > > > > **Thank you for the response**
> > > > > >
> > > > > > Thank you for your detailed response with additional results for different architectures on CIFAR 100. After going through the response and other reviewers' comments, I have decided to keep my score.
> > > > > >
> > > > > > I believe that the paper would greatly benefit from a detailed discussion on the precise notion of memorization the authors aim to study. It would be a very interesting contribution if the authors could make a formal connection with the definition of memorization by Feldman & Zhang. Similarly, as other reviewers have suggested, a theoretical argument that connects the proposed metric with a formal definition of memorization and generalization would enhance the contribution of this paper.
> > > > > >
> > > > > > In response to my question about the novelty of the efficient loss curvature calculation, the authors have responded _"...We found that their method did not work out of the box, and we had to make significant changes to establish a good metric, in essence **reverting back to Hutchinson’s trace estimator form** for calculating curvature [Hutchinson1]."_ Please note that both **Moosavi-Dezfooli et al. (2019)** and Garg & Roy (2023) utilize the **trace of the square of the Hessian matrix** to compute the curvature. Also, Moosavi-Dezfooli et al. (2019) utilize **random directions** to compute curvatures. I agree that the objective of this paper might be different from these prior works and the authors do make interesting contributions regarding averaging loss curvature throughout the training (as already mentioned in my review). However, my earlier comment about **limited novelty** still holds.

---

### Official Review · Reviewer_spXg · 2023-11-01

**Soundness:** 3 good
**Presentation:** 3 good
**Contribution:** 2 fair
**Rating:** 5
**Confidence:** 4

**Summary:**

This paper proposes to use the per-instance curvature metric to calculate the memorization of the sample by a neural network. The curvature is approximated by the trace of the hessian matrix, which is approximated by using Hutchinson’s method. The curvature scores obtained from the proposed method, are shown to correlate well with the scores obtained by Zhang and Feldman. Furthermore, the scores have been shown to be able to detect the samples which have presence of label noise in a synthetic setup.

**Strengths:**

The paper is clearly written and is easy to read.

The proposed curvature score is easier to compute in comparison with Zhang and Feldman, and can be practically used.

**Weaknesses:**

Novelty: The proposed method has a high degree of overlap with Garg and Roy 2023 [R2], where the similar per-instance score is used for determining the curvature. Further, the score in their paper is able to find long-tailed and rare samples, which is the application demonstrated in this paper. It would be great if authors can please clarify the novelty of this work in comparison with Garg and Roy 2023 [R2].

Missing Baselines: The authors don’t compare the proposed method to the Maini et al. 2022 [R1] for identification of the noisy labelled samples, however the setup followed is similar to that of the Maini et al. 2022 paper. I request the authors to please provide the comparison or the reason for the omission of the comparison.

Missing Concrete Application: The section on curvature dynamics for training provides interesting insights. However I couldn’t find any specific experiments which demonstrate its practical applicability.

[R1] Maini, Pratyush, et al. "Characterizing datapoints via second-split forgetting." Advances in Neural Information Processing Systems 35 (2022): 30044-30057.

[R2] Garg, Isha, and Kaushik Roy. "Samples With Low Loss Curvature Improve Data Efficiency." Proceedings of the IEEE/CVF Conference on Computer Vision and Pattern Recognition. 2023.

**Questions:**

I am curious if can the above method be used to identify the samples which have inconsistent captions for the Vision Language Based Methods?

Further, it’s claimed in the paper that FZ scores can’t be used to find samples that are duplicate images with different samples. Can you please provide a concrete reason for this?

Is it possible to provide any theoretical results regarding the correctness of the curvature scores for finding the noisy labeled samples etc?

---

> ### Author Response · Authors · 2023-11-23
> **Response to Reviewer Part 1/3**
>
> ### Response to Reviewer Part 1/3
>
> We thank the reviewer for their time and consideration, and answer the questions here:
> &nbsp;
>
>
> 1. Novelty: The proposed method has a high degree of overlap with Garg and Roy 2023 [R2], where the similar per instance score is used for determining the curvature. Further, the score in their paper is able to and long-tailed and rare samples, which is the application demonstrated in this paper. It would be great if authors can please clarify the novelty of this work in comparison with Garg and Roy 2023 [R2].
>
>     Authors: Thank you for your question. Garg and Roy [R2] indeed was a source of inspiration for this work. We found their analysis of curvature for creating coresets interesting, and were inspired to see if we could use it as a measure of memorization. We found that their method did not work out of the box, and we had to make significant changes to establish a good metric, in essence reverting back to Hutchinson’s trace estimator form for calculating curvature [Hutchinson1]. Their scores also failed to capture the failure mode of duplicated samples with differing labels. We would like to highlight the following differences:
>
>     * In this manuscript, we focus on using curvature as a method for measuring memorization of  samples, and focus on the high curvature samples. In contrast, Garg and Roy 2023 [R2] focus on low curvature samples, and their application to data efficiency for coreset creation.
>
>     * We note that their method of calculating curvature does not give reliable results:
>         *   They used the adversarial direction to estimate curvature. We removed the adversarial direction assumption and returned to traditional Hutchinson’s trace estimator form, and used random Rademacher vectors instead for more reliable results. The cosine similarity with FZ scores on CIFAR100 for Garg and Roy [R2] is 0.17, while our method achieves 0.82.
>
>         * To further improve estimation, we average our results over 10 different random Rademacher variables while [R2] use a single direction. We get a good match for n=5,10,20 with FZ scores as seen in the hyperparameter tuning results from Table 3 in Appendix B.
>
>         * Their method only calculates curvature at the end of training, and the following table shows that that gives unreliable scores. For instance, the match with FZ scores for curvature calculated at the end of training, as shown in the table below, is only 0.18.  We average curvature during training to get reliable results, in order to allow for different decision boundaries that have been learned at different epochs as training progresses. This is the result of studying curvature dynamics during training in Section 4.4 and is quite evident if we form a GIF of the decision boundary in the input space. We have added that to the supplementary material, also found [here](https://imgur.com/a/IGDqwo0). We can see from the first few epochs in the GIF, that the network is trying out different hypotheses in the form of different decision boundaries, on the way to the final one. Averaging over all these hypotheses allows us to be robust to any particular one and shows much more reliable results.
>
>         * We show the results for using Garg and Roy’s method, our method limited to calculating curvature at end of training, and our method as outlined in the paper and show that we get considerably better matches (better cosine similarity CS) with FZ scores for both the most memorized samples, and for the entire dataset, both with (wd1) and without (wd0) weight decay.
>
>
>             |       Method            |      Top 5K FZ CS (wd0)      |   Top 5K FZ CS (wd1)        |      CS with FZ for all data (wd0) |     CS with FZ for all data (wd1)      |
>             |-------------------------|------------------------------|-----------------------------|------------------------------------|----------------------------------------|
>             |      Garg and Roy [R2]  |     0.07                     |     0.24                    |     0.10                           |     0.17                               |
>             |      Ours n=10 EOT      |     0.06                     |     0.28                    |     0.12                           |     0.18                               |
>             |      Ours               |     0.82                     |     0.90                    |     0.73                           |     0.82                               |
>
>         * We provide insights and strong empirical evidence for the use of high curvature samples in studying memorization of deep neural nets, and utilize it to identify a new failure mode that all other methods, including [R2], fails to capture, as shown in Figure 19 in the Appendix.
>
>     We thank the reviewer for helping us consolidate the differences, and have summarized this discussion in the section in Appendix D4.

---

> > ### Author Response · Authors · 2023-11-23
> > **Continued Response: Part 2/3**
> >
> > 2. Missing Baselines: The authors don’t compare the proposed method to the Maini et al. 2022 [R1] for identification of the noisy labelled samples, however the setup followed is similar to that of the Maini et al. 2022 paper. I request the authors to please provide the comparison or the reason for the omission of the comparison.
> >
> >     Authors: We wish to emphasize that Maini et. al is more focused on identifying typical, mislabeled, complex and rare samples, whereas our work focuses on measuring memorization, and use the application of mislabeled sample detection only as a corroboration of our metric. However, we feel that the reviewer is justified in asking for this baseline for the mislabeled sample detection application, as it is a very strong baseline. Hence, we have added it to the paper, using the open-source code available from the author's github repo. From Table 2, we can see that Curvature outperforms SSFT for both MNIST and CIFAR-10 for all percentages of mislabeling considered. However, SSFT performs better than Curvature for CIFAR-100, but as the authors note in their text, SSFT can be combined with other methodologies. We show that combining SSFT and Curvature results in a combination that outperforms both SSFT and Curvature independently and significantly. Additionally, we also note that SSFT results shown in the paper fail to capture the novel failure mode we caught, that of duplicate samples with difffering labels.
> >
> >     &nbsp;
> >
> >     We thank the reviewer for pointing us to a strong baseline, and for helping us strengthen our paper by improving our results by combining it with SSFT. However, we would still like to reemphasize that our paper focuses on establishing curvature as a method for measuring memorization, and the application of detecting mislabeled samples is shown merely as corroboration that curvature does indeed detect memorization.
> >
> >     |  Dataset |   Method  | Corruption |         |         |        |        |        |
> >     |:--------:|:---------:|:----------:|---------|---------|--------|--------|--------|
> >     |          |           | 1%         | 2%      | 4%      | 6%     | 8%     | 10%    |
> >     | MNSIT    | Inconf.   | 99.4\%     | 99.0\%  | 98.4\%  | 97.7\% | 97.1\% | 96.1\% |
> >     |          | CL        | 99.7\%     | 99.3\%  | 99.1\%  | 98.9\% | 98.9\% | 98.9\% |
> >     |          | SSFT      | 99.9\%     | 99.9\%  | 99.8\%  | 99.7\% | 99.7\% | 99.5\% |
> >     |          | LT        | 98.7\%     | 99.5\%  | 98.4\%  | 98.5\% | 97.7\% | 97.3\% |
> >     |          | Curv      | 100.0\%    | 100.0\% | 99.9\%  | 99.9\% | 99.9\% | 99.9\% |
> >     | CIFAR10  | Inconf.   | 84.2\%     | 82.5\%  | 81.8\%  | 81.6\% | 81.5\% | 81.5\% |
> >     |          | CL        | 92.4\%     | 93.1\%  | 93.4\%  | 93.4\% | 91.7\% | 93.4\% |
> >     |          | SSFT      | 94.5\%     | 94.1\%  | 93.2\%  | 92.5\% | 91.6\% | 90.0\% |
> >     |          | LT        | 87.3\%     | 82.5\%  | 84.0\%  | 83.4\% | 82.2\% | 83.0\% |
> >     |          | Curv      | 97.4\%     | 96.6\%  | 95.5\%  | 94.4\% | 94.1\% | 92.9\% |
> >     |          | Curv SSFT | 97.5\%     | 96.8\%  | 96.2\%  | 96.1\% | 95.2\% | 94.8\% |
> >     | CIFAR100 | Inconf.   | 85.0\%     | 84.0\%  | 83.5\%  | 83.6\% | 83.6\% | 83.4\% |
> >     |          | CL        | 83.1\%     | 84.2\%  | 85.3\%  | 86.3\% | 84.3\% | 84.6\% |
> >     |          | SSFT      | 94.8\%     | 93.7\%  | 93.2\%  | 91.7\% | 91.9\% | 91.0\% |
> >     |          | LT        | 85.2\%     | 85.2\%  | 81.8\%  | 80.8\% | 79.8\% | 78.5\% |
> >     |          | Curv      | 90.9\%     | 89.6\%  | 88.3\%  | 86.8\% | 85.7\% | 84.3\% |
> >     |          | Curv SSFT | 96.0\%     | 94.5\%  | 94.2\%  | 93.1\% | 92.9\% | 91.9\% |

---

> ### Author Response · Authors · 2023-11-23
> **Continued Reponse Part 3/3**
>
> 3. Missing Concrete Application: The section on curvature dynamics for training provides interesting insights. However I couldn’t find any specific experiments which demonstrate its practical applicability.
>
>     Authors: Thank you for pointing out the insights in Section 4.4. The major reason for writing section 4.4 was to help the readers to understand why we need to average our scores every epoch as opposed to say, calculating curvature at the end of training. In that way, it serves as the base that informs our design of metric for measuring memorization. We realize that this was not coming across from our writing, and thank the reviewer for identifying the need for a better explanation. We have cleaned up Section 4.4 considerably, and believe it reads a lot better now. We have reduced the content to highlight the following key takeaway, and believe that it is now a lot easier for the reader to follow.
>
>
>     &nbsp;
>     Takeaway: At the early epochs of training, the boundary changes very rapidly and significantly. In fact, every epoch, the boundary can curve around a different subset of samples. This means that measuring the curvature at 1 epoch only will be unstable. Averaging the curvature over all of training helps us iron out this effect and get reliable curvature results. This is quite evident if we form a GIF of the decision boundary in the input space, and we have added that to the supplementary material, also found [here](https://imgur.com/a/IGDqwo0). We can see from the first few epochs in the GIF, that the network is trying out different hypotheses in the form of different decision boundaries, on the way to the final one. Averaging over all these hypotheses allows us to be robust to any particular one and shows much more reliable results.
>
>
> &nbsp;
>
> 4. I am curious if can the above method be used to identify the samples which have inconsistent captions for the Vision Language Based Methods?
>
>     Authors: We believe this is possibly a very good application of our method we have not yet tried. We focused on vision datasets in this paper, and hope that this would encourage others to try our method in different domains.
> &nbsp;
>
> 5. Further, it’s claimed in the paper that FZ scores can’t be used to find samples that are duplicate images with different samples. Can you please provide a concrete reason for this?
>
>     Authors: We wish to clarify that we did not mean that FZ scores cannot calculate this, just that the FZ scores that have been released do fail to catch duplicate samples with different labels. We have edited the paper to make this clear. We cannot recreate FZ scores to try and improve their methodology to catch this failure mode since it requires training 20,000 models. However, we can think of some reasons why they may have failed to capture it. FZ scores are calculated by looking at the change in expected probability of correct prediction when the sample is removed from the training set. To make this computationally easier, the models perform subsampling of the dataset, and it is possible that the considered subsets missed the duplicate samples. It is also possible that each of the 20,000 models was not trained to full convergence for efficiency reasons.
> &nbsp;
>
> 6. Is it possible to provide any theoretical results regarding the correctness of the curvature scores for finding the noisy labeled samples etc?
>
>     Authors: In this manuscript, we use qualitative and quantitative results to show that curvature is a good measure of memorization. We are working on more theoretical findings as part of future work.
>
> &nbsp;
>
> References
>
> Hutchinson1 - Michael F Hutchinson. A stochastic estimator of the trace of the influence matrix for Laplacian smoothing splines. Communications in Statistics-Simulation and Computation, 19(2):433–450, 1990.
>
> R2 - Isha Garg and Kaushik Roy. Samples with low loss curvature improve data efficiency. In Proceedings of the IEEE/CVF Conference on Computer Vision and Pattern Recognition, pp. 20290–20300, 2023.

---

> > ### Comment · Reviewer_spXg · 2023-12-03
> > **Response to the Authors**
> >
> > Thank you for the detailed response and the efforts made in the rebuttal. I find that the changes made have significantly improved the paper.
> >
> > There are still some questions which I feel aren't answered to my knowledge yet, elucidated below:
> >
> > 1. Why does the memorization prediction require averaging across all epochs? While the decision boundary stays similar for many later epochs, as displayed in the gif. It would be great if the authors could analyze this more.
> >
> > 2. The connection of the theoretical definition of memorization to the formulation proposed in this paper can significantly improve the paper.
> >
> > 3. Further, the main novelty is claimed to be in the formulation, which is very similar to prior work [R2,  Moosavi-Dezfooli et al. (2019)], which makes the claim weaker. I would suggest authors look more deeply at the averaging process across epochs, and elucidate the differences more clearly from the prior works.
> >
> > Due to the above concerns, I will currently maintain my rating. However, I will request the authors to take the feedback positively and improve the further versions of the draft.

---

### Official Review · Reviewer_hiPb · 2023-11-08

**Soundness:** 3 good
**Presentation:** 3 good
**Contribution:** 3 good
**Rating:** 6
**Confidence:** 4

**Summary:**

This paper proposes a metric to identify memorized points. The proposed method utilizes the average curvature of the loss function with respect to the input and argues that memorized points have a higher curvature score. They demonstrate this by adding label noise to the training dataset and training until overfitting, showing a higher curvature score for these points during training. They also illustrate quite a high cosine similarity between their score and the previous method by Feldman & Zhang (2020), which they call the FZ score in the paper. However, the advantage of their method is that they don't have to train as many models as required by the FZ method, demonstrating computational benefits.

**Strengths:**

The paper has an acceptable visualization of the experiments; however, more work can be done to make them more accurate and understandable. They have tried to justify their observations and conclusions by designing experiments. Different kinds of datasets are used in this paper, each of which helps to better understand the paper.

In general, I am leaning towards accepting the paper, and I am even open to reconsider my initial score upon improving the quality of the paper regarding the presentation and clarification of questions and points mentioned in the following.

**Weaknesses:**

1. Toy example in section 4.4: The number of training points is far less than in the test set, which is not a similar case to your image datasets. The training data does not have enough points to accurately represent the underlying distribution. Additionally, adding noise to it makes it more challenging for the model to learn the decision boundary correctly.
2. Section 4.4: It is a lengthy section and difficult to follow the main points of it; the presentation format can be improved.
3. Figure 4: It would be better to display mislabeled examples side by side.
4. Figure 5: The labels on the axes of the figure are not clear, and I can't interpret the results. Having a distribution over the corrupted samples would be more informative.
5. Tables: MEAN+STD missing in the tables.
6. Appendix F: It does not explain any correlation between memorized sampel by FZ and your method.

**Questions:**

1. Introduction (Regarding “weakly labeled or have noisy annotations”): Can you apply your method in unsupervised settings?
3. Table 2: Did you use weight decay for other methods as well? Can you show the result of FZ method?
4. Section 4.3: Can you elaborate on why ROC curves are more reliable?
5. How is the threshold for the curvature of loss selected?
6. Does averaging the curvature over epochs remove the specific epoch signals? Isn't it better to study the difference between curvature evolution during time than taking an average?
7. Figure 8: Which type of sample triggers the peak in curvature, and do memorized points contribute more to that?
8. Figure 8: Why is the curvature of test points much higher than memorized points in CIFAR-100? And why don't we see that in ImageNet?
9. End of page 8: What does the sensitivity of test samples to perturbation mean?
10. Have any studies explored using the curvature of samples for inference attacks?

---

> ### Author Response · Authors · 2023-11-23
> **Response Part 1/3**
>
> ### Response to Reviewer, Part 1/3
> Thank you for your encouragement, and for mentioning your willingness to reconsider your review. We address each of the questions below:
>
> &nbsp;
> 1. Regarding the toy example formulation, and the clarity of section 4.4:
>     Question: Toy example in section 4.4: The number of training points is far less than in the test set, which is not a similar case to your image datasets. The training data does not have enough points to accurately represent the underlying distribution. Additionally, adding noise to it makes it more challenging for the model to learn the decision boundary correctly.
>
>     Authors: The imbalance in the design of this toy example is intentional. Our goal was to evaluate what the decision boundary looks like when networks memorize the data. The few, noisy training data encourages the network to memorize the training data. But, having clean abundant test data allows us to also measure generalization of the underlying true data distribution under extreme memorization and noisy samples. This setup allowed us to understand how curvature behaves with extreme memorization, with the following key takeaway:
>
>     At the early epochs of training, the boundary changes very rapidly and significantly. In fact, every epoch, the boundary can curve around a different subset of samples. This means that measuring the curvature at 1 epoch only will be unstable. Averaging the curvature over all of training helps us iron out this effect and get reliable curvature results. This is quite evident if we form a GIF of the decision boundary in the input space, and we have added that to the supplementary material, also found [here](https://imgur.com/a/IGDqwo0). Please look at the early epochs from the GIF, we can see that the neural network is trying out different decision boundaries to try to minimize the loss, since multiple different boundaries give good results. Averaging over these different hypotheses was necessary to not rely too much on one particuaar hypothesis.
>
>     We thank the reviewer for identifying the need for a better explanation, and we have cleaned up Section 4.4 considerably, and believe it reads a lot better now. We have reduced the content to highlight this key takeaway, and believe that it is now a lot easier for the reader to follow.
> &nbsp;
>
>
> 2. Figure 4: It would be better to display mislabeled examples side by side.
>
>     Authors: Thank you for the suggestion, we have made this change to Figure 4.
> &nbsp;
>
> 3. Figure 5: The labels on the axes of the figure are not clear, and I can't interpret the results.
>
>     Authors: Thank you for identifying this readability issue, we have increased the font size and improved the readability of graphs in Figure 5
> &nbsp;
>
> 4. Having a distribution over the corrupted samples would be more informative.
>
>     Authors: We have added a density plot visualizing the distribution of input loss curvature for clean and mislabeled samples in the appendix (See Figure 19).
> &nbsp;
>
> 5. Appendix F: It does not explain any correlation between memorized sample by FZ and your method.
>
>     Authors: We show the quantitative comparison of our score with FZ scores using a cosine similarity metric in Table 1. However, we use Appendix F to emphasize a different point. We display the most memorized samples as per FZ scores, in order to highlight that despite being 3 orders of magnitude more computationally expensive than our method, they do not find the failure case of duplicated samples with differing labels that we found with our analysis. This is significant since the duplicate samples are most definitely memorized by a model if the model gets close to a 100% training accuracy. We highlighted this better in the text in Appendix F.
> &nbsp;
>
> 6. Introduction (Regarding “weakly labeled or have noisy annotations”): Can you apply your method in unsupervised settings?
>
>     Authors: One advantage of our method is that computing curvature of sample only requires a loss, as opposed to other methods that require logits. This renders our method scalable towards unsupervised settings, unlike SSFT or FZ scores. Thank you for highlighting an advantage of our method.
> &nbsp;
>
> 7. Table 2: Did you use weight decay for other methods as well? Can you show the result of FZ method?
>
>     Authors: Yes, all results in Table 2 are using weight decay. For FZ scores however, we were limited to using the results the posted on their GitHub repository as it is not possible to recreate their experiments as they calculated their scores by training 20,000 models per dataset. For this reason, it is also not possible to add that as a baseline to the mislabeled sample detection in Table 2.

---

> ### Author Response · Authors · 2023-11-23
> **Continued Response Part 2/3**
>
> 8. Section 4.3: Can you elaborate on why ROC curves are more reliable?
>
>     Authors: Metrics such as accuracy are very brittle to imbalance in datasets. For instance, if we had a dataset with very few samples of the positive class, such as in our case with mislabeled samples, predicting all negative would give us a hundred percent accuracy, despite nothing having been learnt. This can be avoided by using AUROC curves. We preferred AUROC curves over PR curves as PR curves over represent the smaller class, in our case the mislabeled class. There can be memorized samples in the dataset that were not synthetically mislabeled as well, and we wanted to give a more holistic comparison. However, we realize that this may be more application and user dependent than we estimated, and upon looking closer, we think that giving a generic recommendation is likely not the best. We have removed this line from the text, and thank the reviewer for taking the time to check the minutiae of our paper.
> &nbsp;
>
>
> 9. How is the threshold for the curvature of loss selected?
>
>     Authors: In the paper we do not specifically calculate a threshold, we report the AUROC (Area Under ROC curve), as is commonly practiced in literature. AUROC/AUC is obtained by varying the threshold. The Area under the ROC curve gives a more holistic view than the ROC metric at some threshold. However, given an ROC plot there are several ways in literature to select a suitable threshold. For example, one method is to choose a threshold that maximizes the true positive rate (TPR) while minimizing false positive rate (FPR) i.e. the threshold that results in the top left point of the ROC curve. Another method chooses a threshold such that FPR is below a certain value or TPR is above a certain value. There are other works that inject known label-altered data and track their metric to identify the threshold. Any of these methods could be used if the user wants a threshold for their use case. However, it is common practice to just report AUROC numbers to show good separability between two classes.
> &nbsp;
>
> 10. Does averaging the curvature over epochs remove the specific epoch signals? Isn't it better to study the difference between curvature evolution during time than taking an average?
>
>     Authors: Thank you for your question, this is exactly what we were hoping to answer with the analysis done in Section 4.4. Different examples are memorized at each epoch as the model makes different boundaries in order to learn the data since there can be many correct boundaries. For instance, the boundaries at epochs, 20, 30 and 40 are different in Figure 7. There is inherent noise in how the network reaches the minima during learning as it samples different minibatches stochastically, which results in different curvature scores for samples at each epoch. Averaging over epochs allows us to average over all the decision boundaries the model looked at during its training and make aggregate decisions, which is more reliable. To support our claim, we can show that curvature scores calculated at the end of training are not good enough, and we revamp Section 4.4 to make this point clearer. We provide a link to a [GIF](https://imgur.com/a/IGDqwo0) to illustrate the development of input space curvature. From the early epochs in this GIF, we can note the different hypotheses tried out by the network, evidenced by changing decision boundaries.
> &nbsp;
>
> 11. Figure 8: Which type of sample triggers the peak in curvature, and do memorized points contribute more to that?
>
>     Authors: Figure 8 plots the average curvature of the training set across epochs. The curvature of all samples peak around the overfitting epoch, as explained in Section 4.4. However, from the same figure,  we can see that the curvature of the 5000 most memorized samples according to an independent measure (FZ scores) remains higher than the average curvature across the dataset. Hence, we can indeed conclude that the peak is contributed to more by the memorized samples than the rest of the dataset.
> &nbsp;
>
> 12. Figure 8: Why is the curvature of test points much higher than memorized points in CIFAR-100? And why don't we see that in ImageNet?
>
>     Authors: We believe this is due to ReNet18 being significantly overparameterized for CIFAR100 compared to ImageNet. Thus, with CIFAR100 the model is able to overfit to the training dataset a lot more than on ImageNet, thus widening the gap between train and test examples.

---

> > ### Author Response · Authors · 2023-11-23
> > **Continued Response, Part 3/3**
> >
> > 13. End of page 8: What does the sensitivity of test samples to perturbation mean?
> >
> >     Authors: Thanks for identifying a source of confusion. We mean the sensitivity of loss to perturbation of input. We mean here, that the margin maximization we see for training samples, which causes the network to get overconfident on train samples, and hence the loss does not change much with perturbation of input, does not happen for test samples. However, this does not mean that the curvature ranking for test samples is incorrect. As shown in Figures 16 and 17 in the Appendix, the high curvature samples on the validation sets are also long-tailed and atypical. We have re-written this section.
> >
> > &nbsp;
> >
> > 14. Have any studies explored using the curvature of samples for inference attacks?
> >
> >     Authors: As far as we are aware, there are no such studies since we are the first to show the link between curvature and memorization. However, we are in the process of exploring the use of loss curvature for membership inference attacks as part of a new work. We hope that this manuscript will encourage more experimentation on ways to use curvature as a tool for analysis of networks.
> >
> >
> > &nbsp;
> > Once again, thank you for your time and consideration, and for giving us such a detailed review with the opportunity to improve our paper.

---

### Author Response · Authors · 2023-11-23
**Thank you to all the reviewers; Summary of major changes**

To all the reviewers,

Thank you so much for your valuable comments. We got some great questions that helped us improve our manuscript, both in experimentation and presentation. Here is a summary of the main changes we have made to the manuscript, also highlighted in the updated draft:
&nbsp;


* We have shortened Section 4.4 and made the purpose and key takeaway of that simpler and clearer.
&nbsp;

* We have added baselines of Learning Time (LT) and SSFT as part of our mislabeled samples check, and show that our method outperforms both of them, along with Confident Learning and Inconfidence scores, for all mislabeled percentages for MNIST and CIFAR10. However, for CIFAR100, we outperform LT but note that SSFT slightly outperforms our method, but we show that SSFT can be combined with our method, resulting in superior performance to both SSFT and Curvature individually. Additionally, we also note that SSFT results shown in the paper fail to capture the novel failure mode we caught, that of duplicate samples with differing labels.
&nbsp;


* We have added results for curvature calculated with different architectures for CIFAR100, and their comparison with FZ scores. The main takeaway of these sets of experiments is that different architectures all catch the mislabeled data, and the cosine similarity between curvature and FZ scores of the 5k most memorized samples remains high, at approximately 0.8.
&nbsp;


* We wish to make a case that our results are worth sharing with the community, particularly since we identify an as of yet unobserved failure case in two of the most common vision datasets used for testing models. Additionally, we have significantly strengthened the results of one of our applications, detecting mislabeled data by performing reviewer requested comparisons. In particular, we outperform a very strong NeurIPS 2022 baseline, SSFT, for 2 out of 3 datasets, and show that curvature utilized with SSFT together outperforms either method significantly. We hope the reviewers share our excitement in wanting to share these insights with the community.

---

### Meta-Review · Area_Chair_X51q · 2023-12-10

**Metareview:**

The paper introduces a per-instance curvature metric for estimating memorization in neural networks, aiming to offer a computationally efficient alternative to existing methods. The positive feedback acknowledges the clarity of the paper, the ease of reading, and the practical utility of the proposed curvature score, which is shown to correlate well with the scores obtained by Zhang and Feldman. The authors successfully demonstrate the ability of their method to identify overfitting-prone examples across multiple image classification benchmarks. However, the positive aspects are overshadowed by critical concerns raised by the reviewers.

One of the primary issues highlighted by the reviewers is the significant overlap with Garg and Roy (2023), a similar per-instance score for determining curvature. The lack of acknowledgment and thorough comparison with this prior work raises concerns about the novelty of the proposed method. Additionally, the paper is criticized for not comparing against Maini et al. 2022 for identifying noisy labeled samples, which is seen as a crucial omission that hinders a comprehensive evaluation of the proposed approach.

Furthermore, the reviewers express reservations about the claimed efficiency of the proposed curvature metric, arguing that the computational benefits stem from prior work, limiting the technical novelty of the contributions. The concerns extend to the practical applicability of the curvature dynamics for training, with reviewers noting a lack of specific experiments demonstrating its effectiveness.

Despite the authors' rebuttal, the negative sentiment prevails among the reviewers. The limited scope of the proposed technique, focusing solely on memorization estimation, raises questions about its broader applicability. Reviewers emphasize the importance of establishing a formal connection with the definition of memorization by Feldman & Zhang, calling for a theoretical argument to strengthen the paper's contribution. The perceived lack of novelty in the formulation, similarities with prior work, and concerns about the efficiency of the proposed metric led to a unanimous recommendation for rejection. Addressing these fundamental concerns, providing thorough comparisons with relevant prior work, and establishing a clearer theoretical foundation could significantly improve the paper's chances of acceptance in future submissions.

**Justification For Why Not Higher Score:**

Four out of 5 reviewers stayed negative even after rebuttal indicating that this paper requires a major revision based on the review queries.

**Justification For Why Not Lower Score:**

N/A

---

### Decision · Program_Chairs · 2024-01-16

Reject